# Layered BiOI single crystals capable of detecting low dose rates of X-rays

Robert A. Jagt[1,13], Ivona Bravić[1,2,13], Lissa Eyre [2,3], Krzysztof Gałkowski[2], Joanna Borowiec[4,5], Kavya Reddy Dudipala[6], Michał Baranowski [7,8], Mateusz Dyksik [7,8], Tim W. J. van de Goor[2], Theo Kreouzis [4], Ming Xiao[1,11], Adrian Bevan [4], Paulina Płochocka [7,8], Samuel D. Stranks[2,9], Felix Deschler[2,3,12], Bartomeu Monserrat [1,2] ✉, Judith L. MacManus-Driscoll [1] ✉ & Robert L. Z. Hoye [6,10] ✉

Detecting low dose rates of X-rays is critical for making safer radiology instruments, but is limited by the absorber materials available. Here, we develop bismuth oxyiodide (BiOI) single crystals into effective X-ray detectors. BiOI features complex lattice dynamics, owing to the ionic character of the lattice and weak van der Waals interactions between layers. Through use of ultrafast spectroscopy, first-principles computations and detailed optical and structural characterisation, we show that photoexcited charge-carriers in BiOI couple to intralayer breathing phonon modes, forming large polarons, thus enabling longer drift lengths for the photoexcited carriers than would be expected if self-trapping occurred. This, combined with the low and stable dark currents and high linear X-ray attenuation coefficients, leads to strong detector performance. High sensitivities reaching $1.1 \times 10^3$ $\mu C\,Gy_{air}^{-1}\,cm^{-2}$ are achieved, and the lowest dose rate directly measured by the detectors was $22\,nGy_{air}\,s^{-1}$. The photophysical principles discussed herein offer new design avenues for novel materials with heavy elements and low-dimensional electronic structures for (opto)electronic applications.

Advanced X-ray imaging techniques, such as X-ray fluoroscopy, have significantly enhanced the quality of medical care[1,2]. Reducing the X-ray dose rate would reduce the harm to patients and enable new applications in X-ray imaging, such as X-ray video techniques[3]. However, the lowest dose rate of X-rays detectable by radiology instruments is currently set by the attenuation materials used in the detectors[3]. The linear attenuation coefficient for X-ray absorption can be written as $\alpha \propto Z^4/E^3$, where $E$ is the energy of the radiation detected, and $Z$ the average atomic number of the attenuation material. Standard Si-based photodiodes are unsuitable for X-ray imaging because of the

[1]Department of Materials Science and Metallurgy, University of Cambridge, 27 Charles Babbage Road, Cambridge CB3 0FS, UK. [2]Department of Physics, Cavendish Laboratory, University of Cambridge, 19 JJ Thomson Avenue, Cambridge CB3 0HE, UK. [3]Walter Schottky Institut, Technische Universität München, Am Coulombwall 4, Garching D-85748, Germany. [4]School of Physical and Chemical Sciences, Queen Mary University London, London E1 4NS, UK. [5]College of Physics, Sichuan University, Chengdu 610064, China. [6]Inorganic Chemistry Laboratory, Department of Chemistry, University of Oxford, South Parks Road, Oxford OX1 3QR, UK. [7]Laboratoire National des Champs Magnétiques Intenses, CNRS-UGA-UPS-INSA, UPR 3228 Toulouse, France. [8]Department of Experimental Physics, Wroclaw University of Science and Technology, Wroclaw, Poland. [9]Department of Chemical Engineering and Biotechnology, University of Cambridge, Philippa Fawcett Drive, Cambridge CB3 0AS, UK. [10]Department of Materials, Imperial College London, Exhibition Road, London SW7 2AZ, UK. [11]Present address: School of Microelectronics Science and Technology, Sun Yat-sen University, Guangdong Province 519082 Zhuhai, China. [12]Present address: Physikalisch-Chemisches-Institut, Universität Heidelberg, Im Neunheimer Feld 229, 69120 Heidelberg, Germany. [13]These authors contributed equally: Robert A. Jagt, Ivona Bravić. ✉e-mail: bm418@cam.ac.uk; jld35@cam.ac.uk; robert.hoye@chem.ox.ac.uk

low atomic number ($Z = 14$), yielding low X-ray attenuation coefficients. Therefore, current flat-panel X-ray detectors primarily utilise either amorphous selenium (a-Se, $Z = 34$) as an absorber material, converting the X-rays directly into an electrical signal, or Tl-doped CsI ($Z_{Cs} = 55$, $Z_I = 53$) as a scintillator, in which incident X-rays are converted to optical photons, which are subsequently collected by a photodetector. The main drawback of using scintillators is that the absorbed light is re-emitted isotropically, reducing the spatial resolution. On the other hand, the limited charge-carrier drift lengths and low $Z$ number of a-Se limit the lowest detectable dose rate (LoDD) achievable.

The ideal X-ray absorber material should have a high linear attenuation coefficient for X-rays (i.e. high effective $Z$ and mass density), a large charge-carrier drift length (i.e. large product of mobility and lifetime, or $\mu\tau$), and a low, stable dark current density (i.e. sufficiently large band gap and low ionic drift). Recently, lead-halide perovskites (LHPs) have shown promising material properties for X-ray detection, such as containing heavy elements (Pb and I) and high $\mu\tau$ products, yielding effective X-ray detectors[4–7]. However, LHPs suffer from ion migration, limiting the range of biases that can be applied to the detectors[4–8]. Furthermore, the large quantities of lead required for X-ray detection substantially exceed the legal limits set across many jurisdictions for consumer electronics products, and therefore limits the range of future applications[9,10].

A promising route to overcoming the aforementioned challenges is to replace lead with the more benign element bismuth[11], such as in the double perovskites (e.g. $Cs_2AgBiBr_6$[8,12]) or $A_3Bi_2X_9$ compounds (e.g. $(NH_4)_3Bi_2I_9$[13–15]). These materials exhibit the important advantages of being more environmentally and thermally stable than LHPs, as well as having higher activation energy barriers to ion migration[13]. However, bismuth-based perovskite-inspired materials have a lower electronic dimensionality than LHPs[16], and experiments have shown that photoexcited carriers in these materials couple strongly to the lattice. In particular, for both $Cs_2AgBiBr_6$ and $A_3B_2X_9$ compounds, which have been the leading Bi-based radiation detectors thus far, the carriers localise in the lattice through self-trapping[17,18]. This severely lowers mobilities and can, in some cases, limit lifetimes, thus limiting the $\mu\tau$ products achievable[17–22]. Understanding how charge-carriers couple to phonons and the strength of this coupling in these systems is important, and finding materials that do not self-trap essential[16].

In this work, we reveal the strong potential of bismuth oxyiodide (BiOI) for X-ray detection. Recent works have suggested BiOI to be a promising visible-light harvester for photovoltaics[23], photoelectrochemical cells[24], and photodetectors[25], but, in the existing literature, the suitability of BiOI for radiation detection is unclear. Previous work on BiOI[26] (and other bismuth chalcohalides[27]) focussed on nanocrystalline detectors, which have low X-ray sensitivities well below $1\,\mu C\,Gy_{air}^{-1}\,cm^{-2}$[26]. To eliminate the deleterious effects of grain boundaries and establish the effectiveness of BiOI as an X-ray detector, we devise a method to grow single crystals with low defect densities. We find that the layered structure of BiOI and large band gap at room temperature (1.93 eV) enable stable and high electrical resistivity in the dark, resulting in low dark currents. The high effective $Z$ number and mass density give rise to strong X-ray attenuation. Through detailed spectroscopic measurements and first-principles calculations, we show that the photoexcited charge-carriers structurally deform the lattice due to coupling to two breathing-mode phonons to form large polarons (where the charge-carrier wavefunction is delocalised over several unit cells) rather than small polarons or self-trapped excitons (where the charge-carrier wavefunction is localised on the order of a unit cell or smaller). Together with the low defect densities in the BiOI single crystals, the delocalised nature of the polarons enable large $\mu\tau$ products[28]. We validate our results by demonstrating devices capable of detecting lower dose rates of X-rays (down to $22\,nGy_{air}s^{-1}$) than MAPbBr$_3$ ($39\,nGy_{air}s^{-1}$, MA = methylammonium)[5], $Cs_2AgBiBr_6$ ($60\,nGy_{air}\,s^{-1}$)[12] and $(NH_4)_3Bi_2I_9$ ($55\,nGy_{air}\,s^{-1}$)[13], and well below the current medical standard of $5500\,nGy_{air}\,s^{-1}$[23].

## Results

### BiOI single crystal growth & materials characterisation

In order to assess the potential of BiOI for X-ray detection it is imperative to understand the nature of the photoexcited carriers, how they interact with phonons, and whether any localisation, that lowers the mobility and limits the detector response, occurs. To elucidate these properties, we evaluated the BiOI single crystals through photoluminescence (PL) and transient absorption (TA) techniques.

BiOI has a layered structure belonging to the space group $P4/nmm$, with lattice parameters $a = b = 3.99$ Å and $c = 9.21$ Å at room temperature, where the stoichiometric I-Bi-O-Bi-I layers are stacked along the $c$ axis (Supplementary Fig. 1, Supplementary Tables 1–3). Owing to this layered structure, BiOI grows anisotropically, with the (00 $l$) face much larger than the thickness (see Methods for more details). We confirmed that the diffraction peaks from this top face only arose from (00 $l$) peaks (Fig. 1a), consistent with its single crystalline nature. The high content of bismuth in BiOI gives it a high effective $Z$ value of 73.6 (see Supplementary Note 1 for calculations) as well as a high mass density, $\rho$, of $7.97\,g\,cm^{-3}$ (Supplementary Table 1). These values for BiOI are larger than other state-of-the-art X-ray detector materials, such as CdZn$_x$Te$_{1-x}$ ($Z \le 50.2$, $\rho = 5.85\,g\,cm^{-3}$), MAPbI$_3$ ($Z = 64.1$, $\rho = 4.5\,g\,cm^{-3}$), $Cs_2AgBiBr_6$ ($Z = 60.0$, $\rho = 4.65\,g\,cm^{-3}$), and $(NH_4)_3Bi_2I_9$ ($Z = 62.1$, $\rho = 4.3\,g\,cm^{-3}$)[4,12,13]. We grew BiOI single crystals using a chemical vapour transport (CVT) method in sealed quartz ampoules[29,30] (see Methods). The crystals grow as platelets due to the layered structure[31].

The temperature-dependent PL spectra obtained when illuminated with a 2.33 eV continuous wave (cw) laser are depicted in Fig. 1b. At room temperature, the PL signal is weak and 200 meV red-shifted with respect to the absorption edge. Upon cooling to 120 K, both the PL intensity, energy shift and lifetimes increased (Fig. 1b, c). Interestingly, the PL spectrum appears to be composed of two emissive states (Fig. 1b), which are energetically separated by 11.9 meV (Supplementary Fig. 4a and Supplementary Note 4)[32]. The shape of these cw-excited PL spectra depends on the excitation wavelength (see Supplementary Fig. 4). When we used a pulsed laser excitation (with a period of 1 ms that allows all photoexcited charge-carriers to decay before the next pulse), we found that the PL shape is independent of the excitation wavelength. Directly after photoexcitation, no change in the PL shape from the original PL spectrum at time $t = 0$ ps was observed (measured with a streak camera, see Supplementary Fig. 5). This indicates that the observed excitation-wavelength dependence of the PL is due to the interaction of the excited state carriers with the incident light source.

We found there to be a rise in the PL intensity upon cooling, along with a simultaneous increase in the PL lifetime, as measured by time-correlated single-photon counting (TCSPC) measurements (Fig. 1c). More specifically, when cooled from room temperature down to 80 K, the PL lifetime (determined by mono-exponential fits) increased from 2 ns to 6.8 μs. We performed temperature-dependent XRD measurements to exclude the possibility of phase transitions over this temperature range (Supplementary Fig. 6). Furthermore, from the increase in the PL intensity with decreasing temperature, we determined the thermal activation energy barrier for the quenching of the PL to be 25.4 meV (Supplementary Note 4). Therefore, at room temperature, thermal energy (26 meV) is sufficient to enable carriers in the photoexcited state to couple to the ground state and relax non-radiatively (details in Supplementary Note 4), limiting the PL lifetime to only 2 ns. It was only by reducing the thermal energy that this non-radiative

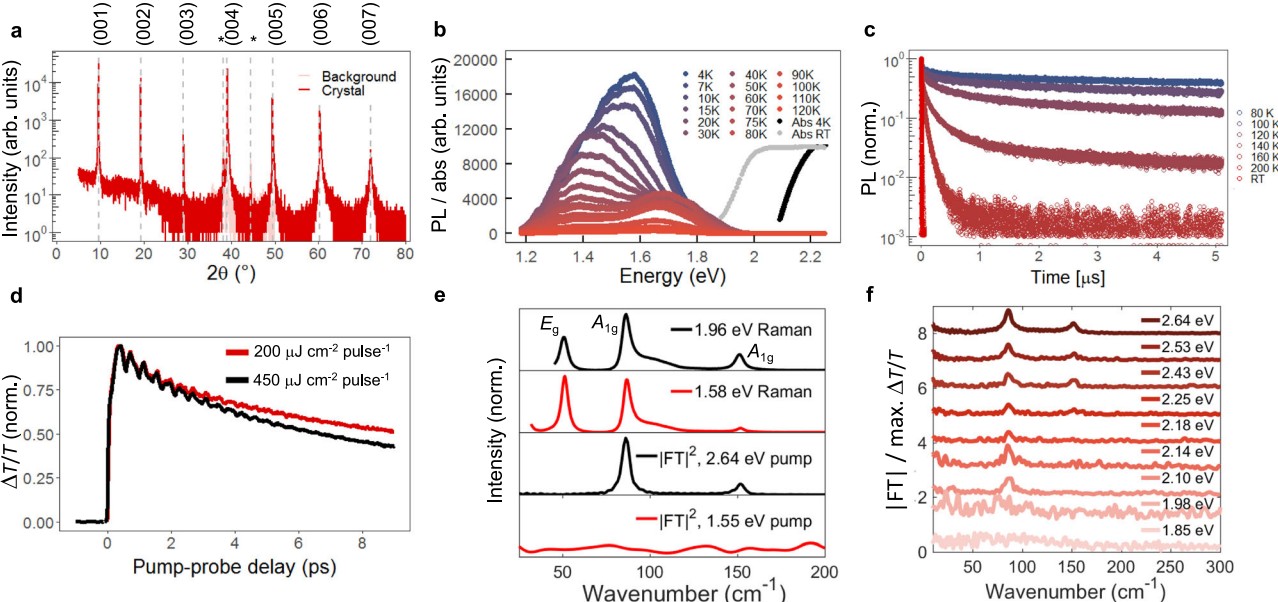

**Fig. 1 | Material properties of BiOI single crystals. a** X-ray diffraction (XRD) pattern of a BiOI single crystal plotted on top of the measured background. The vertical lines correspond to the (00 *l*) reflections of the BiOI XRD pattern. **b** Temperature-dependent photoluminescence (PL) spectra of BiOI single crystals (532 nm wavelength cw excitation) from 4 K (dark blue) to 120 K (bright red), together with the absorption spectra (Abs) at 4 K (black) and room temperature (grey). **c** Decay of the PL over time, measured using time-correlated single photon counting (TCSPC) from 80 K (dark blue) to room temperature (bright red). **d** Normalised differential transmission ($\Delta T/T$) of BiOI thin films after a 2.64 eV pump pulse as a function of a 2.07 eV probe delay, depicted for a 200 μJ cm$^{-2}$ pulse$^{-1}$

(red) and 450 μJ cm$^{-2}$ pulse$^{-1}$ (black) pump fluence. **e** Normalised resonant (black) and non-resonant (red) Raman spectra and Fourier transformed transient absorption (TA) signal using an above-bandgap pump (2.64 eV, 450 μJ cm$^{-2}$ pulse$^{-1}$) and below-bandgap pump (1.55 eV, 1 mJ cm$^{-2}$ pulse$^{-1}$). Probe energy for both TA plots was 2.08 eV. All spectra were normalised to the peak at 86 cm$^{-1}$ except the below-bandgap pump TA, which was normalised to the peak at 0 cm$^{-1}$. **f,** Fourier transform (FT) spectra of the oscillations in differential transmission at a probe energy of 2.08 eV and excited at a range of pump energies (1.85 eV to 2.64 eV). Each spectrum is normalised to the maximum $\Delta T/T$.

energy relaxation process could become less prominent, such that the PL lifetime substantially increases.

The large full width at half maximum (FWHM -0.4 eV) of the PL peaks, along with their large energy shift (-0.7 eV) from the optical band gap indicate that the photoexcited carriers decay in a deformed lattice. Both features would typically be attributed to the formation of self-trapped excitons[18–20,22,32–34], which would be detrimental for transport lengths. We, therefore, delved deeper to understand whether localisation through self-trapping occurs in BiOI.

To elucidate the mechanism behind the change in PL spectra with temperature, we performed transient absorption (TA) measurements of BiOI thin films on glass substrates[35]. The film was excited with a 2.64 eV energy pulsed laser and the kinetics of the ground state bleach, measured at two pump fluences, are shown in Fig. 1d. In the first few picoseconds, a beating pattern is observed, which scales with the fluence of the laser, indicating that the oscillations are caused by the presence of photoexcited carriers. After performing a Fourier transform on the oscillating component (see Supplementary Note 5), two modes were found to contribute to the coherent oscillations – an 86 cm$^{-1}$ (11 meV) mode and a 152 cm$^{-1}$ (19 meV) mode (Fig. 1e). We found there to be close agreement between the frequencies of these modes and the peaks observed from the Raman spectra (Fig. 1e). Both peaks were assigned to symmetry-retaining, intralayer breathing modes of $A_{1g}$ symmetry (shown in Fig. 2d). However, the 50 cm$^{-1}$ mode found from Raman spectra measurements does not appear in the TA kinetics. The transient oscillations in the pump-probe signal are due to impulsive absorption, which generate coherent phonons and distort the lattice along the two intralayer breathing modes after photoexcitation[22,36–38] (see Supplementary Note 5 for further discussion). The spectrum of coherent oscillations shows no shift in the frequencies of the phonon modes with decreasing pump energy, and the peak amplitudes do not show a strong trend, when normalised to

the peak $\Delta T/T$ value (Fig. 1f). However, at pump energies below 2.25 eV, the 150 cm$^{-1}$ (19 meV) peak disappears, and at pump energies below 1.98 eV (close to the indirect band gap), neither peak could be resolved. Therefore, the excess energy of photoexcited carriers directly affects which phonon modes are populated by impulsive absorption.

Therefore, Fig. 1 shows that photoexcited charge-carriers in BiOI structurally distort the lattice along the intralayer breathing modes and, from this observation, we propose that the large red-shift in the PL peaks to the optical absorption spectrum is predominantly caused by the displacive excitation of coherent phonons (DECP), rather than a self-trapping process. Under continuous illumination, this structural deformation can be controlled by the excitation wavelength. Finally, we note that we compared the PL decay of BiOI crystals with 400 nm and 530 nm wavelength excitation, and found no significant change in the PL lifetime (Supplementary Fig. 4g). This observation suggests that there are no dangling bonds that potentially create surface in-gap states at the probed (00 *l*) surface (see diagrams in Supplementary Fig. 1), and that the measurements made and discussed here are reflective of the bulk properties of BiOI.

## First-principles studies

To test the hypothesis that the relaxation of the BiOI lattice due to photoexcited carriers occurs predominantly along the dominant longitudinal optical phonons (instead of localised distortions), we performed state-of-the-art electronic structure calculations. Our framework combines $GW$[39] and Bethe-Salpeter calculations[39–41] for the excited-state properties with a frozen-phonon approach to capture the renormalisation of the photoexcited state with respect to the displacement amplitude along the aforementioned Raman modes. Importantly, this many-body formalism allows us to describe any scenario between an exciton with a large binding energy and strong

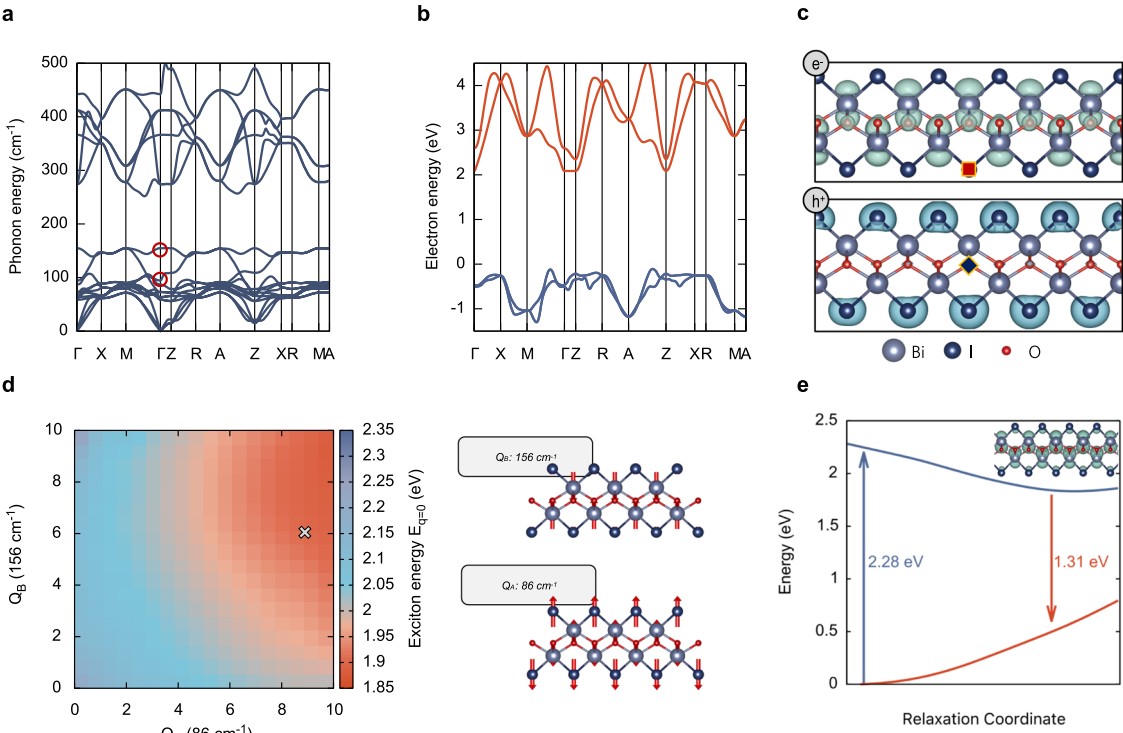

**Fig. 2 | First principles studies of BiOI. a**, Phonon dispersion curve of bulk BiOI. Red circles indicate the two $A_{1g}$ phonon modes $Q_A$ and $Q_B$. **b** Fully relativistic single-shot *GW* band structure of BiOI. The highest occupied state is set to 0 eV **c** Top: spatial distribution of the electron constituting the lowest-lying direct exciton when the hole (red square) is fixed on an iodine atom. Bottom: spatial distribution of the hole constituting the lowest-lying direct exciton when the electron (blue square) is localised on an oxygen atom. **d** Two-dimensional cut of the potential energy surface corresponding to the lowest-lying direct exciton. The axes correspond to the positive displacements along the Raman active modes illustrated on the right-hand side, with the minimum labelled by a grey cross. $Q_A = 10$ corresponds

to the collective displacement of all I atoms out of the layer by c = 0.63 Å, while $Q_B = 10$ corresponds to the displacement of Bi atoms into the layer by c = 0.27 Å. **e** Relaxation coordinate diagram illustrating the lowering of the exciton energy upon distortion along Raman modes $Q_A$ and $Q_B$. It is evident that the renormalisation (stabilisation) of the excited state (upon distortion) is much smaller than the destabilisation of the ground state. The energy minimum of the excited state is at $Q_A = 9$, $Q_B = 6$. Inset: Electron density of the excited state after structural displacement along the relaxation coordinate highlights that the wavefunction remains fully delocalised. The corresponding hole wavefunction is depicted in the Supplementary Fig. 31.

localisation, all the way to uncorrelated photoexcited charge-carriers, and we, therefore, avoid simplified descriptions of self-trapped excitons or large polarons. By solving the excitonic eigenvalue problem explicitly through the Bethe-Salpeter equation, we are also not reliant on approximate models of the exciton. Through these calculations, we indeed confirm that the coherent vibration of both oscillations with $A_{1g}$ symmetry is causing the giant red-shift in the PL spectrum whilst retaining the delocalised character of the excited state due to conservation of the crystal symmetry.

We used density functional theory (DFT)[42,43] to calculate the ground state structural properties of BiOI, and performed said calculations with the electronic structure code QUANTUM ESPRESSO, in this case without the inclusion of spin-orbit coupling (SOC). We obtained static-lattice parameters of $a = b = 3.97$ Å and $c = 9.17$ Å, which are in good agreement with the experimentally-determined low-temperature lattice parameters (Supplementary Fig. 8a, b). From the finite-displacement method in conjunction with nondiagonal supercells (see details in Methods) we constructed the phonon-dispersion curve, which is illustrated in Fig. 2a. The phonon modes of interest are the Raman modes resolved in the TA experiment and are highlighted by the red circles in the phonon-dispersion curve. Both Raman modes correspond to longitudinal $A_{1g}$ out-of-plane breathing modes, one mode where the iodine (and, with smaller amplitudes, bismuth) atoms move parallel to the out-of-plane direction (Bi-I bond vibration), and another mode where the bismuth atoms move in the same fashion (Bi–I and Bi–O bond vibration) as illustrated in Fig. 2d, right.

We subsequently calculated the single-shot *GW* band structure including SOC with the YAMBO package, which is depicted in Fig. 2b, and identify the indirect nature of the fundamental band gap, in accordance with previous first-principles studies[28]. The fundamental quasiparticle (QP) band gap has a static lattice value of 2.27 eV, while the first direct transition has a value of 2.33 eV (which lies at the Z-point of the electronic Brillouin zone). The associated in-plane electron and hole effective masses at the band edges are estimated to be 0.23 and 0.26, respectively. When we accounted for electron-hole correlation, which is obtained via solving the Bethe-Salpeter equation, the indirect band gap and first direct transition become 2.24 eV and 2.28 eV, respectively. We estimated through computations the static exciton binding energy (of the undistorted system) for the indirect exciton to be 35 meV, and for the direct exciton to be 45 meV. These binding energies are surprisingly low for a layered van der Waals material which is a result of the large interlayer thickness. The calculated eigenvalues are in excellent agreement with the onset of the low-temperature absorption spectrum (~2.2 eV) in Fig. 1b.

Due to the same parity of the valence and the conduction band representations at the Z-point, the vertical transition across the first direct transition is symmetry forbidden and thus dark, leading to an excited state with a long lifetime at cryogenic temperatures. However, the flatness of both the valence and conduction bands along the Z → Γ high-symmetry line enables the admixture of multiple electron-hole pairs in proximity to the optical band gap, which creates an excited state with a non-zero transition probability. The corresponding electron and hole densities of the excitonic wavefunction are illustrated in

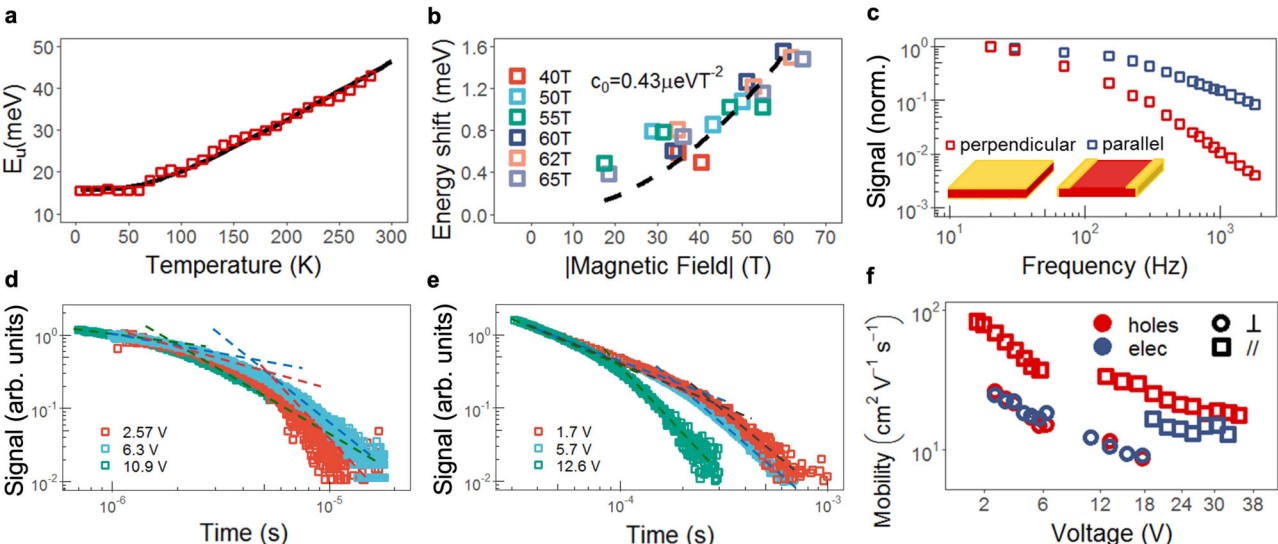

**Fig. 3 | Photophysical properties in BiOI. a** The Urbach energy ($E_U$) as a function of temperature, determined from transmission experiments. **b** Measured change in absorption edge as a function of magnetic field strength (colours correspond to different applied fields, key shown inset). Dashed line indicates parabolic fit through the data, with fitted scaling constant $c_0$ as depicted. **c** Normalised photocurrent obtained using a lockin-amplifier (excitation from 3.06 eV cw laser) as a function of chopping frequency for both the parallel (dark blue) and perpendicular devices (red). Device structures inset. **d**, **e** Normalised transient current curves of the BiOI single crystal device under various biases for the (**d**) perpendicular and the (**e**) parallel device structures. Colours correspond to the applied biases, which are shown in the key inset in each sub-plot. Tangents fit to obtain the arrival times are indicated. Please refer to Supplementary Fig. 14 for more detailed views of these tangents. **f** Time-of-flight carrier mobility of holes (red) and electrons (dark blue) as a function of applied bias for both the perpendicular (circular symbols) and parallel (square symbols) direction.

Fig. 2c. Figure 2c is a simple representation of the six-dimensional exciton wavefunction, in which we either show the electron wavefunction when the hole is fixed to an iodine atom (top panel), or the hole wavefunction when the electron is fixed to an oxygen atom (bottom panel). The electron wavefunction (Fig. 2c, top) resembles the conduction band at the Z-point which is predominantly made of antibonding-type hybridisation of Bi $6p_z$ orbitals, with a small admixture of I $5p_z$. The hole, on the other hand, resembles the valence band at the Z-point which is generated from hybridisation of nonbonding iodine 5p states and thus is confined in the iodine sublayers. The photoexcitation process can therefore be described as a charge-transfer like state from the outer iodine sublayers to the inner Bi−O−Bi sublayer.

Having understood the excited state, we then derived the mechanism which leads to the spectral changes in PL with temperature. We hypothesised that the generation of excitons, and the resulting redistribution of charge-carriers, facilitates a structural reorganisation similar to the displacive excitation of coherent phonons (DECP) of the Raman modes measured from TA. To test this, we took advantage of the results from the transient absorption experiment and reorganised the system along the identified $A_{1g}$ Raman modes with energies of 86 cm$^{-1}$ (11 meV) and 156 cm$^{-1}$ (19 meV), which was achieved by distorting the atomic positions along the individual phonon modes[44] we calculated in the finite-displacement method ($Q_A$ and $Q_B$) to determine the phonon-dispersion curve (details in Methods). The maximum displacement amplitudes of $Q_A$, $Q_B = 10$ corresponds to the collective displacement of all iodine atoms out of the layer by $c = 0.63$ Å ($Q_A$), or all bismuth atoms into the layer by $c = 0.27$ Å ($Q_B$), which are on the same order as the expected thermal displacement of iodine and bismuth from equilibrium from X-ray diffraction measurements[45]. We calculated the ground- and excited-state surface relative to the ground-state equilibrium energy, and the resulting contour plot of the exciton potential energy surface (PES) is depicted in Fig. 2d. Illumination of the system with energies larger than the band gap leads to a charge redistribution between the sublayers (out-of-plane direction), which in turn facilitates out-of-plane displacements.

Upon distortion we find that the superposition of both oscillations causes a strong renormalisation of the first excited state (resulting from a strong quasiparticle gap renormalisation and a lowered exciton binding energy, with the former dominating) into an extremely flat region of the excited state PES. We found that the minimum of the two-dimensional surface (highlighted with a cross in Fig. 2d) has an excitation value of 1.31 eV, which is in excellent agreement with the lowest-lying PL peak at room temperature. At this configuration the exciton binding energy of the direct exciton is lowered from 45 meV to 16 meV, which results from a combination of the significantly increased electron-hole distance and the increased dielectric screening due to the larger layer thickness (from 6.2 Å to 6.8 Å), as well as the decreased interlayer distance (from 3.3 Å to 2.3 Å). In contrast to the excited-state PES, which is extremely flat in proximity to the minimum, the ground state energy surface remains very steep. The different curvatures of the ground and excited state (Fig. 2e) give rise to the broad PL peaks. The electron and the hole densities of the excited state equilibrium show that the exciton-(LO)phonon coupling retains the delocalisation (i.e., wavefunctions spread over multiple unit cells) of the photo-excited state in the in-plane direction and does not facilitate the self-trapping of the exciton (see Fig. 2e inset, with details in Supplementary Fig. 31, and experimental verification later on in Fig. 3b). This can be easily understood considering the symmetry and the displacement vectors of the excited state and the phonons, respectively. The charge densities of the electron and the hole are fully symmetric and entirely delocalised in the in-plane directions, leading to quasi-two-dimensional charge-carriers that are confined within the I−O−Bi−O−I sublayers. The coherent oscillations caused by photoinduced charge redistribution are fully symmetric vibrations (with lattice periodicity) along the c-direction, perpendicular to the extended wavefunctions. Thus, the vibrations along the coherent modes do not trap the charge-carriers in the I−O−Bi−O−I -sublayer. The new structure and the associated translational invariant wavefunctions are illustrated in Supplementary Fig. 31. Our detailed analysis, therefore, shows that the broad and red-shifted PL peak can be fully explained by delocalised polarons, rather than self-trapping.

It is also worth highlighting that the coherent displacements lead to an excitonic band inversion between the direct and indirect exciton which is why we conclude that the PL arises from the direct exciton (a more detailed discussion in Supplementary Note 10).

Altogether, from the first-principles calculations, we obtain that the photoexcited carriers distort the lattice along the two intralayer breathing modes (due to the intermediate Fröhlich coupling and soft lattice) and that the carrier wavefunction retains its delocalised character. These findings, in combination with the photoluminescence experiments above, suggest that excitation at the band edges creates delocalised charge-carriers. Other phonon modes, apart from these two intralayer modes, are of course present, as shown in the phonon dispersion curve in Fig. 2a. Calculating the effects of charge-carrier interactions with all of these modes is computationally prohibitive because the number of possible phonon modes is very large, and capturing all of these phonon modes within finite differences would require very large supercells that would make the BSE calculations extremely computationally expensive. Experimentally finding from TA measurements (Fig. 1d–f) that two phonon modes dominate coupling to charge-carriers was therefore critical to making an advanced first-principles analysis of this system computationally tractable.

We finally add that we do not rule out the possibility that the stereochemistry of the $6s^2$ lone pair of $Bi^{3+}$ could play a role on local distortions in the crystal structure. Whilst this is worth a detailed follow-on study, it is not apparent that the $6s^2$ lone pair is responsible for the phonon modes created after photo-excitation (see Supplementary Fig. 7 and 8), and any static off-centering of atoms, if they do occur, would not account for the changes in the lattice observed after photo-excitation.

## Delocalised nature of photoexcited carriers in BiOI

The delocalised nature of the wavefunction of the carriers and the intermediate in-plane Fröhlich coupling ($\alpha$ in the range of 1.17–1.68, see Supplementary Note 13) to the LO phonons suggests that the carrier mobility is not substantially reduced upon photoexcitation.

The combined effect of the intermediate Fröhlich coupling and soft lattice can be probed by measuring the sub-band gap absorption, which is characterised by the Urbach energy (see Supplementary Note 6)[46]. We determined the temperature-dependent Urbach energy (depicted in Fig. 3a) from optical transmittance measurements (see Supplementary Note 2). For the principal interacting phonon mode, we obtained an energy of 18.8 meV (in excellent agreement with the 19 meV $A_{1g}$ mode), and an estimated steepness parameter of $\hat{\sigma}_0 = 0.58$. No vertical offset in Urbach energy was needed to account for structural disorder, highlighting the crystals to be of high quality, and that the Urbach edge arises due to dynamic rather than static disorder[47]. The low steepness parameter indicates that the carriers strongly distort the ionic lattice[33].

However, despite this structural distortion, the in-plane wavefunction remains delocalised. To probe the radial extent of the exciton we performed transmission experiments at 2 K in the presence of strong magnetic fields up to 65 T (see Supplementary Note 7 for details). At these temperatures we have excitons rather than free carriers, which follows from the linear dependence of the PL on excitation power (Supplementary Fig. 9) and confirmed by first principles calculations. By determining the shift in the absorption edge as a function of the magnetic field, we fitted the coefficient for the diamagnetic shift and inferred the radial extent of the 1s exciton in the undistorted system in the in-plane direction. In this work, a diamagnetic shift coefficient $c_0$ of $0.43\ \mu eVT^{-2}$ was obtained (Fig. 3b), yielding a r.m.s. radius of the 1s exciton of $r_{r.m.s.} = \sqrt{8m_r c_0}/e = 15.3$ Å, where $m_r = 0.12$ is the in-plane reduced mass obtained from first principles calculations. These values are similar to other layered materials (e.g. $WS_2$), and further support a picture of 2D Wannier-type excitons spanning several unit cells in-plane[48].

To study the transport properties, we created two device architectures using semi-transparent Au contacts: one in which the electric field is perpendicular to the I–Bi–O–Bi–I sublayers, and one in where it is parallel to these planes (see Fig. 3c, inset). The BiOI single crystals are highly resistive, with $\rho_{perpendicular} = 1.1 \times 10^{12}\ \Omega\,cm$ and $\rho_{parallel} = 1.8 \times 10^9\ \Omega\,cm$ (Supplementary Fig. 10), due to the large band gap (1.93 eV) and near intrinsic Fermi-level (see Supplementary Fig. 11). The resistivity in the perpendicular configuration is orders of magnitude larger than the parallel configuration because of suppressed ion migration (see Supplementary Fig. 33 and 34). That is, in the in-plane direction, mixed ionic and electronic transport occurs to a much larger extent than in the out-of-plane direction, resulting in an orders of magnitude difference in resistivity despite the mobilities in the two directions being similar. Notably, the resistivities obtained from BiOI in the perpendicular configuration is orders of magnitude larger than the resistivities of recently-reported novel bismuth-halide semiconductors (e.g., $MA_3Bi_2I_9$; $10^{10}-10^{11}\ \Omega\,cm$)[13,14], CZT ($10^{10}\ \Omega\,cm$)[14] and 3D perovskites ($10^7-10^8\ \Omega\,cm$)[14]. We measured the frequency response of the photocurrent upon chopped cw laser excitation (3.06 eV) with a lock-in amplifier. The frequency (at 2.1 V bias) at which the drop in signal was −3 dB was 265 Hz for the parallel device and 63 Hz for the perpendicular device (Fig. 3c).

To determine the perpendicular and parallel carrier mobilities, we performed time of flight (ToF) measurements. Selective illumination through one of the Au contacts and changing the bias polarity facilitates the measurement of both the electron and hole mobility. The resulting photocurrent, as a function of laser pulse delay time, is depicted for the perpendicular (Fig. 3d) and parallel structures (Fig. 3e – see Supplementary Fig. 13). From the arrival time present in the ToF signal (please refer to Supplementary Fig. 14 for more detailed views), we determined the charge carrier mobility using the relation $\mu = \frac{L^2}{Vt}$ where $L$ is the distance between the electrodes, $V$ the applied bias, and $t$ the arrival time. The voltage-dependent mobility for electrons and holes is depicted in Fig. 3f. The maximum perpendicular (26 $cm^2\,V^{-1}\,s^{-1}$) and parallel mobilities (83 $cm^2\,V^{-1}\,s^{-1}$) were obtained at a low bias voltage of 2 V, with similar values for electrons and holes, which is consistent with the similarity between the electron and hole effective masses (0.23 and 0.26, respectively). The parallel mobility agrees well with the mobility of 54 $cm^2\,V^{-1}\,s^{-1}$ obtained from space-charge limited current density (SCLC) measurements, from which we observe a clear quadratic space-charge regime in the parallel device configuration (Supplementary Fig. 12). From SCLC measurements, we obtained a low trap density of $2.3 \times 10^9\ cm^{-3}$, which is similar to or lower than other state-of-the-art X-ray detection materials (e.g., $Cs_2AgBiBr_6$[13], $Cs_3Bi_2I_9$[16], $MAPbI_3$[32]), emphasising the high quality of the BiOI crystals.

## BiOI X-ray detectors

The delocalised nature of the photoexcited carrier wavefunction, fast photo-response, and low and stable dark currents make BiOI a highly promising candidate for X-ray detectors. Furthermore, the linear X-ray attenuation coefficient of BiOI surpasses many popular materials classes used for X-ray imaging, such as lead-halide perovskites, $CdZn_xTe_{1-x}$ (e.g. CdTe), and double perovskites (Fig. 4a). We took radiographs of BiOI single crystals in a side-by-side comparison with silicon, and showed that far fewer X-rays were transmitted (only 2%, compared to 78% for silicon), demonstrating the higher stopping power of BiOI (Fig. 4a, inset). Only 134 µm thick BiOI crystals are needed to attenuate 90% of 30 keV X-ray photons, whereas $MAPbI_3$ would require 360 µm thick crystals (Supplementary Fig. 15). To validate the potential of BiOI as an X-ray detector, we focus on devices made in the perpendicular configuration (Fig. 4c, inset), owing to lower dark currents that are stable under applied bias (Supplementary Fig. 10). Furthermore, the perpendicular devices had no ionic currents over the entire range of fields tested (up to 1667 V $cm^{-1}$; Supplementary Fig. 33). We determined the X-ray detection properties by measuring the

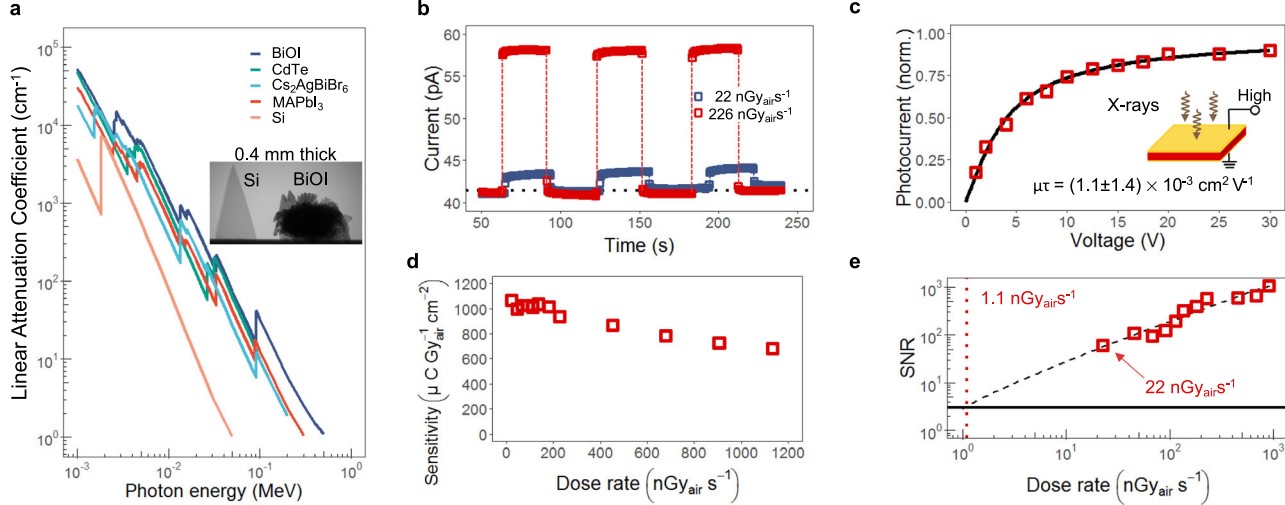

**Fig. 4 | X-ray detection properties of BiOI single crystals and performance of devices in the perpendicular configuration. a** Linear attenuation coefficients as a function of photon energy for BiOI (dark blue), CdTe (green), $Cs_2AgBiBr_6$ (light blue), $MAPbI_3$ (red) and Si (pink). Inset are radiographs of Si (0.4 mm thick) and BiOI crystals stacked together to give the same thickness of 0.4 mm. The transmittance through Si is 78%, whereas through BiOI, it is 2%, demonstrating the high stopping power of BiOI. **b** Photocurrent measurement of the perpendicular BiOI devices at X-ray dose rates of 22 $nGy_{air}$ $s^{-1}$ (dark blue) and 226 $nGy_{air}$ $s^{-1}$ (red). **c** Photocurrent of perpendicular BiOI devices illuminated with X-rays (square symbols) and fit with

the modified Hecht equation (black line) to extract the mobility-lifetime ($\mu\tau$) product (see Supplementary Fig. 17 for photo- and dark current values). Device structure inset. **d** Measured X-ray sensitivity of the perpendicular devices as a function of dose rate. **e** X-ray dose rate-dependent signal-to-noise ratio (SNR) of the perpendicular device structure. Solid black line represents SNR = 3. Dashed black line is a linear fit to the measured SNR as a function of X-ray dose rates (square symbols), and the dashed red line shows the dose rate (1.1 $nGy_{air}$ $s^{-1}$) at which this linear fit intersects with SNR = 3. For parts **b**, **d** and **e** the applied field was 278 V cm$^{-1}$ (5 V applied bias).

photo-response upon X-ray illumination (Fig. 4b), and the details of the setup are in Supplementary Fig. 16. For the X-ray source we used a Cu anode X-ray tube, with a maximum X-ray energy of 35 keV and peak intensity at 7.9 keV (see Supplementary Fig. 16d), with the dose rate calibrated using an X-ray ion chamber dose meter. These energies are slightly lower than the energies used in medical X-ray tubes (typically between 20 and 150 keV). From the photoresponse, the $\mu\tau$ product can be determined by fitting the modified Hecht equation[12,13]:

$$I = \frac{I_0 \mu\tau V}{L^2} \frac{1 - \exp\left(\frac{-L^2}{\mu\tau V}\right)}{1 + \frac{Ls}{V\mu}} \qquad (1)$$

where $I_0$ is the saturated photocurrent, $s$ is the surface recombination velocity and $L$ and $V$ are the distance between the electrodes and the applied voltage, respectively. The effective mobility-lifetime product obtained was: $\mu\tau_{\text{perpendicular}} = (1.1 \pm 1.4) \times 10^{-3}$ cm$^2$ V$^{-1}$ (Fig. 4c; see also Supplementary Fig. 17). We found that the photoconductivity was not dependent on the polarity of the applied electric field, indicating that the electron and hole $\mu\tau$ products are similar. Similar behaviour was found in the parallel configuration, and this detailed in Supplementary Note 14, from which we obtained a higher mobility-lifetime product of $(6 \pm 2) \times 10^{-2}$ cm$^2$ V$^{-1}$. Nevertheless, devices made in the perpendicular configuration have higher charge-collection efficiencies for the same applied bias, due to the shorter transport distances and higher electric fields (see Supplementary Table 4). The sensitivities of $(1.1 \pm 0.1) \times 10^3$ µC $Gy_{air}^{-1}$ cm$^{-2}$ (Fig. 4d) reached in the perpendicular configuration substantially exceeds the performance of previously-reported nanocrystalline BiOI detectors[26]. To determine the LoDD, we measured the signal-to-noise ratio (SNR) of the X-ray detectors down to 22 $nGy_{air}$ s$^{-1}$, at which point the SNR was 61, well above the IUPAC standard for the LoDD (SNR of 3; Fig. 4e). We were not able to provide lower dose rates due to instrument limitations. But following approaches taken in the literature[14], linearly extrapolating the dose rate values down to an SNR of 3 gives an intersection at 1.1 $nGy_{air}$s$^{-1}$,

which shows that the LoDD values can be very low in this material. Furthermore, the lowest dose rate directly measured in these BiOI single crystals (22 $nGy_{air}$ s$^{-1}$) is lower than the LoDDs of novel metal-halide X-ray detectors, such as $MAPbBr_3$ (36 $nGy_{air}$ s$^{-1}$)[4], $Cs_2AgBiBr_6$ (59 $nGy_{air}$ s$^{-1}$)[12], $(NH_4)_3Bi_2I_9$ (55 $nGy_{air}$ s$^{-1}$)[13], $Cs_3Bi_2I_9$ (130 $nGy_{air}$ s$^{-1}$)[15], and comparable to the best value reported for $MA_3Bi_2I_9$ (6.5 $nGy_{air}$ s$^{-1}$ directly measured, 0.62 $nGy_{air}$s$^{-1}$ extrapolated)[14], as well as being several orders of magnitude lower than what is needed for regular medical diagnostics (5500 $nGy_{air}$ s$^{-1}$)[15]. A full comparison of the key properties and performance of the BiOI detectors with other semiconductor detectors is given in Supplementary Table 5.

## Discussion

Beyond its strong performance as an X-ray detector, this work on BiOI single crystals puts forward important insights into the development of materials containing heavy metal elements for radiation detection, as well as more broadly for optoelectronics. Typically, the observation of a broad red-shifted PL is ascribed to self-trapped excitons, as has been widely found among bismuth-based perovskite-inspired materials[16,32]. In contrast, we show that in BiOI, this phenomenon can be entirely explained by Fröhlich interactions. The resulting large polarons are delocalised and exhibit band-like transport, thus enabling higher mobilities and faster detector responses than if small polarons or self-trapped excitons formed. We show that this structural reorganisation of the lattice is an intrinsic property of the material, rather than being induced by crystal defects. This suggests that the structural deformation can be controlled by strategic materials design (e.g., by changing the number of layers or composition), which can lead to further compounds with similar desirable properties. In contrast to typical van der Waals materials, each layer in BiOI is comprised of a larger number of atoms through the cross-section (i.e., I–O–Bi–O–I), which results in the material exhibiting two breathing modes (instead of one) that modify the excited state. We propose that layered ionic materials (where the charge densities of the electrons and holes are fully symmetric and entirely delocalised in the in-plane directions)

composed of heavy atoms and exhibiting multiple breathing modes (i.e., contain multiple layers), will exhibit similar properties to BiOI. In principle, the symmetry and delocalisation of the wavefunctions can be easily checked by examining the projected density of states of candidate materials, in which the conduction band is primarily composed of states of the inner atoms (e.g., Bi and O) and the valence band primarily outer atom states (e.g., I).

Another important insight from this work is that the reorganisation of the lattice from carrier-phonon coupling can result in an unavoidable non-radiative loss channel, even if the interactions are only Fröhlich. This loss channel limits the PL lifetime in BiOI to 2 ns at room temperature, thus limiting the out-of-plane diffusion length to ~0.4 μm, well below the 180 μm transit distances in the perpendicular devices. The thin films used in optoelectronics will have shorter diffusion lengths, thus highlighting the limitations of using BiOI for diffusion-driven devices, such as photovoltaics. Nevertheless, we show that applying an electric field allows these carriers to be efficiently extracted. For example, applying a small bias of 1.8 V across the electrodes of the perpendicular devices produces an electric field of 100 V cm$^{-1}$. As a result, a drift length on the order of 1 mm would occur, thus allowing efficient charge-carrier extraction. At the same time, dark current densities remain low (1 pA mm$^{-2}$ at 1.8 V, or 13 pA mm$^{-2}$ at 5 V). The long drift lengths and the low dark current densities together enable the high sensitivities and low LoDDs we observed in X-ray detectors. We attribute the differences between drift and diffusion lengths to be due to the applied field causing charge-carriers to be decoupled from lattice distortions, and therefore avoid the non-radiative loss channel due to Fröhlich coupling, such that much longer drift lifetimes are obtained. Indeed, we could see from the temperature-dependent time-resolved PL measurements (Fig. 1c) that suppressing this loss channel by depopulating the dominant phonon modes allows orders of magnitude increases in the PL lifetime. This motivates a reconsideration of the applications of perovskite-inspired materials containing heavy elements (which have currently mostly focussed on photovoltaics), since we show that high-performance can still be achieved in drift-driven devices (such as X-ray detectors), even if there is significant carrier-phonon coupling that limit diffusion lengths[8,12–15,20,21,34,35].

In summary, we showed that photoexcited charge-carriers in BiOI couple to two optical phonon intra-layer breathing modes. The low stiffness of the lattice along the out-of-plane (c-axis) direction, along with the intermediate coupling between carriers and longitudinal optical phonons, results in a structural deformation of the lattice along the c-axis. The changes in the lattice energy in the excited and ground state with these distortions causes the PL to become red-shifted, and the PL lifetime fundamentally limited to approximately 2 ns at room temperature due to the creation of a non-radiative loss channel. We show that this limitation can be overcome by i) depopulating the dominant intra-layer phonon modes, resulting in PL lifetimes increasing to the microsecond level at 80 K, or ii) applying an electric field to decouple charge-carriers from lattice vibrations, thus enabling long drift lengths on the millimeter-scale at room temperature. Furthermore, the excited carrier wavefunctions remain delocalised, yielding highly mobile carriers with large drift $\mu\tau$ products. The BiOI single crystals have a high electrical resistivity, and high linear X-ray attenuation coefficients, making them ideal X-ray detector materials. The optimised devices exhibited a high sensitivity of up to $(1.1 \pm 0.1) \times 10^3$ μC Gy$_{air}^{-1}$ cm$^{-2}$ and a low detection limit of 22 nGy$_{air}$ s$^{-1}$ due to the high drift $\mu\tau$ products, and low dark current densities on the order of pA mm$^{-2}$. These findings emphasise drift-driven devices to be an important application route for perovskite-inspired materials. Importantly, our work offers structural and electronic insights into how carrier localisation could be avoided, forming a basis for the future design of perovskite-inspired materials with delocalised excitations and long charge-carrier transport lengths.

## Methods

### Materials
BiI$_3$ (Sigma Aldrich, 99.99% metals basis), Bi$_2$O$_3$ (Sigma Aldrich, 99.99% metals basis), deionised water (as produced), Au pallets (Kurt Lesker, 99.99%) were used without further purification.

### Growth of BiOI single crystals
BiI$_3$ (0.5897 g) and Bi$_2$O$_3$ (0.46596 g) are pressed in equimolar amounts into a pellet, together with 18 μL of deionised water, which acts as a transport agent. This pallet is placed inside an ampoule, evacuated to a rough vacuum (0.1 mbar) and refilled with argon (3 times), before it is evacuated to a high vacuum (10$^{-4}$ mbar), and subsequently sealed using a gas flame torch. The ampoule is place inside a two zone furnace, with the pallet kept at 720 °C and the crystal growth zone at 680 °C for ~5 days, with an initial heating rate of 1 °C min$^{-1}$ for both zones. In order to achieve low bulk defect densities, the crystals needed to be cooled very slowly (<0.1 °C min$^{-1}$, total cooling time ~5 days) to prevent the defects formed at elevated temperatures from being "frozen in" when the cooling rate is too fast (see Supplementary Note 3). Due to its anisotropic structure, BiOI crystals grow as platelets, with significantly larger in-plane than out-of-plane (i.e., along crystal thickness) size[31]. The typical dimensions of the crystals were 1–5 mm width/length, and 0.05–0.2 mm thickness, with the flat reflecting surface corresponding to the (00 *l*) plane, which is evident from the X-ray diffraction (XRD) pattern (Fig. 1a; Supplementary Figs. 2 and 3). Clear optical interference fringes in the sub-band gap transmittance were visible, with the maximum transmittance reaching 100%, indicating high crystal surface quality and smoothness (see Supplementary Note 2). The exact dimensions of the crystals used for the data depicted in Figs. 3 & 4 are shown in Supplementary Fig. 10.

### Device fabrication
After growth the BiOI single crystals were rinsed with isopropanol and transferred to a high vacuum chamber of a thermal evaporator and dried for 2 h. A 30 nm semi-transparent gold contact was evaporated using a shadow mask.

### Materials characterisation
X-ray diffraction was performed using a Panalytical Empyrean D8 theta/theta system. Cu K$_\alpha$ radiation (λ = 1.5406 Å) was used as the X-ray source.

X-ray photoemission spectroscopy was undertaken by a monochromatic Al K$_\alpha$ X-ray source (hν = 1486.6 eV) using a SPECS PHOIBOS 150 electron energy analyser with a total energy resolution of 0.5 eV. The measurements were performed at 300 K.

For the measurements below using pulsed excitation sources, the fluences used are specified in the captions of the figures they relate to.

Time-resolved photoluminescence measurements and PL measurements with long interval times (1 ms) were performed by exciting the samples using a 400 nm wavelength frequency doubled Ti:Sapphire laser (Spectra Physics Solstice). The repetition rate was 1 kHz and pulse length approximately 100 fs. The PL spectra were measured at 5 ns time intervals using an intensified charge-coupled device camera with an Andor iStar DH740 CCI-010 system connected to a grating spectrometer (Andor SR303i).

Further time-resolved photoluminescence (TRPL) measurements were carried out using a confocal microscope setup (PicoQuant, MicroTime 200). The excitation laser, a 532 nm pulsed diode (PDL 828, PicoQuant, pulse width of around 100 ps), was directly focused onto the sample with an air objective. The emission signal was separated from the excitation light using a dichroic mirror. A pinhole of 50 μm was included in the detection path, as well as an additional 550 nm longpass filter to minimise the laser contribution to the recorded signal. The TRPL was then focused onto a Hybrid PMT detector connected to a PicoQuant acquisition card for time-correlated single

photon counting (time resolution of 100 ps). The repetition rates were set to 10 MHz.

PL and PLE spectra were obtained using a fixed grating Maya 2000 Pro spectrometer. A He flow vacuum cryostat was used to regulate the temperature.

Ultraviolet–visible spectroscopy was conducted on a home-built setup, utilising a tungsten white light source and diffraction grating to measure the transmittance. The BiOI crystals were mounted inside a vacuum-cryostat (base pressure $10^{-4}$ mbar) onto a helium cooled cold finger with 1 mm diameter openings.

Transient absorption spectroscopy measurements were performed using the same laser source as for the photoluminescence measurements. The source laser was also frequency-doubled to give a 400 nm wavelength pump beam. To generate a broadband (500–800 nm wavelength) probe beam, a home-built noncollinear optical parametric amplifier was used. The sample was aligned such that the pump and probe beams overlapped. Adjacent to these was a reference probe beam to account for shot-to-shot variation in probe transmission through the sample. The imaging spectrometer was an Andor Shamrock SR303i and the detector a pair of linear image sensors (Hamamatsu, G11608). To determine the differential transmittance ($\Delta T/T$), a mechanical/electronic chopper to create the "pump on" and "pump off" periods for each measurement, detected with a lock-in amplifier, was used.

Magneto-transmission measurements were performed in pulsed magnetic fields up to 68 T with a pulse duration of ~500 ms. The broadband white light was provided by a tungsten halogen lamp. The magnetic field measurements were performed in the Faraday configuration with the c-axis of the sample parallel to the magnetic field[49]. The light was sent to the sample by an optical fibre. The transmitted signal was collected by a lens and coupled to another fibre. The signal was dispersed by a monochromator and detected using a liquid-nitrogen-cooled CCD camera. The sample was placed in a liquid helium cryostat. Temperature-dependent transmission measurements were performed in the same setup. In such a configuration, the probed area of the sample was in the range of a few hundred $\mu m^2$. Further details can be found in Ref. 49.

Radiographs of the single crystals were taken using a Zeiss Versa 610 X-ray computed tomography (CT) scanner. X-rays were generated using a 3 W power source and 40 kV voltage. The exposure time was 120 s.

## First-principles calculations

The detailed flow chart of our computational methodology is given in Supplementary Fig. 35. We performed the ground state geometry optimisation of BiOI using the planewave pseudopotential implementation of density functional theory (DFT)[42,43] as provided by the electronic structure software package QUANTUM ESPRESSO[50,51] and cross-checked with VASP[52–55]. We used the exchange-correlation functional Perdew-Burke-Ernzerhof revised for solids (PBEsol[56,57]), employed an energy cut-off of 60 Ry and used a $12 \times 12 \times 4$ **k**-point for sampling the electronic Brillouin zone (BZ). The structure was constrained to its initial symmetry throughout the geometry optimisation, and convergence was achieved with forces below $10^{-3}$ eV Å$^{-1}$ and stress components below $10^{-2}$ GPa. Starting from the optimised geometry we performed phonon calculations using the finite displacement method[58] in conjunction with nondiagonal supercells[59], with the same computational parameters as above, and with commensurate **k**-point grids for the sampling of the electronic Brillouin zone of the non-diagonal supercells. The Hessian of each nondiagonal supercell was calculated through the displacement of each atom from its equilibrium position by 0.01 Å in symmetry-inequivalent directions and calculation of the force constants by using finite differences. The constructed matrix was then Fourier transformed to the dynamical matrix and diagonalised to obtain the vibrational frequencies and eigenvectors.

We obtained the Born effective charges and the macroscopic dielectric tensor using the electronic structure code VASP, and subsequently calculated the irreducible representations of the Raman modes with Phonopy[60]. The static and high frequency dielectric constants computed were 43.33 and 8.68, respectively (refer to Supplementary note 13). We found that both Raman modes discussed in the manuscript belong to the same $A_{1g}$ irreducible representation, which is in contrast to prior assumptions in the previous literature, that instead assigned the second Raman mode (156 cm$^{-1}$) to the $E_g$ irreducible representation, but are in-line with published first-principles calculations[61].

In order to determine the quasiparticle $GW$[39] band structure we first calculated the single-particle Kohn-Sham trial wavefunctions from QUANTUM ESPRESSO, followed by a single-shot $GW$ calculation as it is implemented in the YAMBO code[62,63]. The DFT calculations were performed with the same exchange correlation functional as described earlier (i.e., PBEsol), but now using the full relativistic norm-conserving pseudopotentials for Bi, O and I generated with the ONCVPSP code by D. R. Hamann[64,65]. For the $GW$ band structure we applied the plasmon–pole approximation for the inverse dielectric matrix[66]. We found that including 1600 Kohn-Sham states in the sum over states in the correlation part of the self-energy, 600 bands to build the RPA response function and an energy cut-off of 10 Ry for the dielectric matrix leads to a converged quasiparticle band gap. For a speed-up of the convergence with respect to empty states we adopted the terminator technique from Bruneval and Gonze[67].

In order to calculate the dielectric response under consideration of excitonic effects, we solved the Bethe-Salpeter equation (BSE[39–41]) as it is implemented in YAMBO. The BSE calculations were performed on top of the single-particle DFT calculation including a scissor shift to correct the electronic band gap according to the quasiparticle correction in the preceding $GW$ step. The BSE was solved within the Tamm-Dancoff approximation[68] (which is generally valid for bulk compounds to describe neutral excitations well below the plasma frequency of the material) and fully taking into account the spinorial nature of the wavefunctions. Eight occupied and six unoccupied states in the excitonic Hamiltonian prove sufficient to fully converge the position of the first exciton peak.

Up to now, all many-body calculations were performed using a $12 \times 12 \times 4$ **k**-point grid to sample the electronic BZ. However, the convergence of the dielectric constant and the exciton-binding energy with respect to the **k**-point grid can be very slow, especially in systems with flat band edges. Therefore, we calculated the absorption spectrum with a $12 \times 12 \times 4$, $18 \times 18 \times 6$ and $24 \times 24 \times 8$ **k**-point grid and observed that the spectral line shape is not fully converged for a $24 \times 24 \times 8$ **k**-point grid (see Supplementary Fig. 24). However, for the scope of this work, the position of the absorption onset and the spectral line shape is sufficiently converged. Using the results from the calculated spectra, we extrapolated the binding energy of the first direct exciton to an infinite **k**-point grid and obtained an estimate for the exciton binding energy (neglecting dynamical effects) of approximately 45 meV.

In order to assess the relative energy position of the Γ and the finite-momentum excitons, we calculated the exciton dispersion curve along the Γ → X high symmetry line and found that the indirect exciton is ~40 meV below the direct exciton. This scenario changes when we distort the system along the Raman modes in the next step. The exciton dispersions for the ground state and the excited state equilibrium geometry are depicted in Supplementary Figs. 29 and 30, respectively.

Using the frozen-phonon approach, we construct one hundred distorted configurations along the Raman modes and plot the contour of the potential energy surface of the first direct excitonic state in Fig. 2d. We then predict the relaxation trajectory and plot the one-dimensional reaction coordinate in Fig. 2e. The maximum

displacement amplitudes of both modes are listed in angstrom in the Supplementary Note 9.

## Device characterisation

The frequency-dependent photoresponse was measured using a lock-in amplifier. The laser system used was a Pico Quant picosecond pulsed diode laser driver PDL 800-D with a LDH405 pulsed laser diode head (405 nm excitation wavelength, pulse length of 20 ps) to appropriately excite above the band gap of the BiOI crystal. A Zurich Instruments HF2LI 50 MHz lock-in amplifier was used for the lock-in amplification of the current signal from the device and the HF2 LabOne software was used for frequency, lock-in signal and phase acquisition. The stage on which the sample was placed, and the confocal microscope was part of a WITec alpha 300 s setup. Further details can be found in Ref. 69.

Current-voltage measurements to determine the SCLC mobility were measured in the dark using Keithley 2623 A source-measure unit.

The optical excitation for the Time of Fight experiments was provided by the 6 ns duration pulses of 532 nm wavelength laser. The DC bias was provided by a low noise power supply and the photo-current transient measured as the voltage drop across a 10–98 kΩ load resistor connected to the input of a Techtronix TDS510A digitising oscilloscope. For the bulk (out-of-plane) measurements, a collimated expanded beam was incident over the whole surface of the (semi-transparent) illuminated electrode with the electric field lines orthogonal to the surfaces defined by the two electrodes. The direction of the electric field defined the polarity of the charge carrier traversing the bulk of the sample, since the penetration depth of the 532 nm illumination is ~100 nm, i.e. much smaller than the sample thickness (183 μm). For the in-plane measurements, the beam was focussed, using a cylindrical lens, into an illuminated line (approximately 20 μm in width) which was aligned parallel and adjacent to the edge of one of the surface electrodes. The polarity of the sampled charge carriers was again defined by the direction of the electric field between the two surface electrodes (~2.1 mm apart), which is parallel to the crystal surface, within the laser line penetration depth. Please refer to Supplementary Fig. 13 for the optical micrograph of an example of a parallel device. The maximum electric field density used in the devices was 164 V mm$^{-1}$ for the perpendicular device and 4.8 V mm$^{-1}$ for the parallel device.

## X-ray detector performance

The X-ray absorption properties were calculated using the online calculation tool called XCOM, made available by the National Institute of Standard and Technology from the U.S. department of commerce, which can be accessed via: https://doi.org/10.18434/T48G6X.

The calibration of the X-ray source was performed using a Mini Ion Chamber Survey Meter model 2130. The ion chamber is yearly calibrated by the radiation specialist of Imperial College London. To ensure conditions were as similar as possible to those experienced by the sample, the ion chamber was mounted in a similar position and aligned with the X-ray source (see Supplementary Fig. 16 for photograph of the setup).

The photoresponse of the devices upon X-ray illumination were measured using an Keithley 2623 A source-measure unit. Under a continuous fixed bias, the photocurrent was measured. The X-ray tube was turned on and off in 30 s intervals. The acceleration voltage was 35 kV and the dose rate was altered using the tube current.

## Data availability

The experimental data that support the plots within this paper and other findings of this study have been deposited in the Research Data Repository at Imperial College London under accession code 12435. The solved crystal structure for BiOI is available from the Cambridge Crystallographic Data Centre (CCDC) under the accession number 2201891, and can be found in: https://www.ccdc.cam.ac.uk/structures/Search?Ccdcid=2201891&DatabaseToSearch=Published.

## Code availability

The code and computational files that support the plots within this paper and other findings of this study can be found in the same repository as the experimental data, that is the Research Data Repository at Imperial College London under accession code 12435.

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

## Acknowledgements

We would like to thank Prof. Richard Phillips (University of Cambridge) for useful feedback on the manuscript and help with optical measurements. The authors also thank Zhuotong (Thomas) Sun (University of Cambridge) for assistance on the powder X-ray diffraction measurements, and Prof. James Marrow and Marcus Williamson (University of Oxford) for assistance in taking radiographs. R.A.J. acknowledges funding from an EPSRC Department Training Partnership studentship (no. EP/N509620/1), as well as Bill Welland and the Winton Programme for the Physics of Sustainability. L.E. and T.V.D.G. acknowledge support from the EPSRC Cambridge NanoDTC (no. EP/L015978/1). L.E. acknowledges funding by the DFG (project no. 387651688). T.V.D.G. also acknowledges financial support from the Schiff Foundation. K.G. and S.D.S. acknowledge the EPSRC (no. EP/R023980/1) for funding. S.D.S. acknowledges the Royal Society and Tata Group (no. UF150033) and EPSRC (no. EP/W004445/1) for funding. The work has received funding from the European Research Council under the European Union's Horizon 2020 research and innovation programme (HYPERION - grant agreement no. 756962; PEROVSCI - 957513). The work was supported by a Royal Society International Exchanges Cost Share award (no. IEC\R2\170108) and the Alliance Hubert Curien Programme of the British Council (no. 608412749). K.R.D. thanks the Department of Chemistry at the University of Oxford for a studentship. P.P. appreciates support from National Science Centre Poland within the OPUS program (no. 2019/33/B/ST3/01915). This work was partially supported by OPEP project, which received funding from the ANR-10-LABX-0037-NEXT. The Polish participation in European Magnetic Field Laboratory is supported by the DIR/WK/2018/07 grant from Ministry of Science and Higher Education, Poland. F.D. acknowledges support from the DFG Emmy Noether Programme (project no. 387651688) and the Winton Programme for the Physics of Sustainability. J.L.M.-D. acknowledges funding from the Royal Academy of Engineering under the Chair in Emerging Technologies Scheme (no. CIET1819_24). R.L.Z.H. acknowledges support from the Royal Academy of Engineering under the Research Fellowship scheme (no. RF\201718\1701), the Isaac Newton Trust (Minute 19.07(d)), Downing College Cambridge through the Kim and Juliana Silverman Research Fellowship, and an EPSRC grant (no. EP/V014498/2). I.B. and B.M. acknowledge support from the Winton Programme for the Physics of Sustainability. B.M. also acknowledges support from a UKRI Future Leaders Fellowship (no. MR/V023926/1) and from the Gianna Angelopoulos Programme for Science, Innovation and Technology. The calculations are conducted using resources provided by the Cambridge Tier-2 system, operated by the University of Cambridge Research Computing Service (www.hpc.cam.ac.uk) and funded by EPSRC Tier-2 capital grant (no. EP/P020259/1).

## Author contributions

R.A.J., I.B., B.M., and R.L.Z.H. designed the project. R.A.J. grew and characterised the BiOI single crystals. J.L.M.-D. contributed to the materials concepts and growth. I.B. performed the computational calculations. L.E. and F.D. performed the TA measurements. K.G., R.A.J. and S.D.S. performed the PL measurements. M.B., M.D., R.A.J., K.G. and P.P. performed the high magnetic field measurements. R.A.J. (with assistance from Richard Phillips) performed the temperature dependent transmission measurements. R.A.J. and T.K. performed the time-of-flight measurements. R.A.J., J.B., T.K. and A.B. performed X-ray detection measurements. T.V.D.G. performed the low temperature X-ray diffraction measurements. M.X. performed the transient current measurements, which R.L.Z.H. analysed to determine the activation energy barrier to ion migration. K.R.D. performed the X-ray transmittance measurements. All authors contributed to writing the paper.

## Competing interests

The authors declare no competing interests.
