## [Peer Review File · Nature Communications]

Layered BiOI single crystals capable of detecting low dose rates of X-raysEditorial Note: Parts of this Peer Review File have been redacted as indicated to maintain the confidentiality of unpublished data.

REVIEWER COMMENTS

Reviewer #1 (Remarks to the Author):

In this manuscript, Jagt et al. report the result from their systematic study of BiOI single crystals for detecting low dose rates of X-rays, which is critical for making safer radiology instruments. Through use of ultrafast spectroscopy, first-principles computations and detailed optical and structural characterisation, they obtained high sensitivity and low dose rate BiOI-based X-rays detecting device. Moreover, they discussed the underlying physical mechanism of these phenomena, and offer new design avenues for novel materials with heavy elements and low-dimensional electronic structures for (opto)electronic applications. This manuscript is excellent in terms of research conception, design, analysis and discussion of results, obtained device performance. So, I think this manuscript is a well-written article containing interesting results which merit publication.

1. After reading this manuscript, I think the outstanding problem is that the analysis and discussion of the results are not deep enough. The authors adopted a very classic writing style: separating the presentation of the results from the discussion. However, in the last section "Discussion", the authors actually provide more analysis of the underlying mechanism. In the previous sections of the results, the authors described the data in detail and well explained the differences between the results under different conditions. This is necessary for in-depth understanding of data and phenomena, but it will also submerge the discussion and presentation of system mechanism and new physical insights.
2. In the first paragraph of Page 15, the conclusion "Calculating the effects of charge-carrier interactions with all of these modes is computationally prohibitive, and we therefore focus only the two modes that we found experimentally from TA analysis (Fig. 1d–f) to dominate coupling to charge-carriers." needs more detailed derivation and discussion.
3. In the determination of crystal structure, the authors grind the prepared single crystal and test it with powder XRD. Why not do single crystal XRD test directly? Isn't it possible to get more accurate results?
4. In the summarizing paragraph, the authors only summarized the results of this research work in general terms, and did not highlight their findings and innovations.
5. In the experimental method, the authors should provide key technical parameters.
6. In the description of DFT calculation method, the authors use various calculation software and methods. Are the results reasonable and reliable?

Reviewer #2 (Remarks to the Author):

The author reported the use of BiOI in X-ray detection, characterized the coupling of electrons and phonons, and through first-principle calculations, systematically analyzed the process of photoexcited carriers causing lattice distortion and forming large polarons, and give the reason for the high $\mu\tau$ value of the material. In addition, a single crystal with low defect density was synthesized, and a radiation detection device with surprisingly excellent performance was made. I think the explanation of the photoexcited charge-carriers is great, but the X-ray detection performance is under debate. The current states of this paper is not suitable for publication in Nature Commun.

1. Firstly, the PL decay time is 2 ns at room temperature, but the $\mu\tau$ product is measured as $10^{-3} \text{ cm}^2 \text{ V}^{-1}$. This indicates a much higher mobility than the measured results ($10^2 \text{ cm}^2 \text{ V}^{-1} \text{ s}^{-1}$). I think the measurements and fitting of $\mu\tau$ requires further study. I recommend the use of alpha particles for excitation, and pulse height spectra to derive the average response amplitude.
2. Moreover, the authors reported the perpendicular device had higher sensitivity but lower $\mu\tau$ product. This phenomenon is abnormal. The higher resistivity and lower dark current cannot

determine the sensitivity, since sensitivity is caused by the carrier collection efficiency. 3. In discussion part, the discussion about the limitation for diffusion-driven devices is not comprehensive. For X-ray detection, the drift length is also very long, the use of external field could not only increase the extraction efficiency, but also increase the dark current. Thus, for X-ray detection, it is also required for long carrier lifetime and diffusion length.

Other questions:

1. In Supplementary Fig. 14g, I suggest that the author use a logarithmic coordinate system instead of the linear coordinate system, just like Fig 1c. Only attenuation to 20% cannot effectively judge the fitting value of the lifetime. At the same time, it is better to choose a longer range of abscissa. Furthermore, it seems that the wavelength of 400 nm has a longer life under excitation, and I'm curious about the reason.

2. I have some confusion about the carrier mobility tests.

(1) In Supplementary Fig. 12, the ohmic zone should be larger, including the part with a smaller slope in the TRAP-SCLC zone. In my opinion, their slopes are very close to 1. This is in sharp contrast to the part with a slope far greater than 1.

(2) Why the SCLC test is only carried out in one direction? It is suggested that the experiment in the vertical direction can be supplemented.

(3) When the resistivity differs by three orders of magnitude, the mobility of parallel electrodes and vertical electrodes is similar. This doesn't seem very reasonable.

(4) In Fig. 3d and Fig. 3e, the inflection point is blocked by the fitting point. It is suggested that these lines can be separated. If the inflection point is unclear, you can add auxiliary lines or supplement the enlarged picture. It is not recommended to use solid big dots for marker symbols.

3. The authors mentioned that the distortion of the lattice could be controlled by material design. Among the materials with broad and Stokes-shifted PL peak, can you give readers more suggestions through first-principle calculations, and screen out materials with the nature of delocalization instead of self-trapping?

Reviewer #3 (Remarks to the Author):

Manuscript titled "Layered BiOI single crystals capable of detecting low dose rates of X-rays" discusses performance of the device with appreciable sensitivities reaching $1.1 \times 10^3 \mu\text{C Gy}^{-1} \text{cm}^{-2}$ are achieved, and the lowest dose rate directly measured by the detectors was 22 nGy^{-1} . Experimental findings discussed in this manuscript is worth for publish it in nature communication because of the novelty, systematic analysis and interpretations.

Queries to Authors:

1. What happen to the attenuation value of BiOI if the single crystal replaced with compacted nanocrystalline BiOI based pellet?

2. In supplementary fig 32 (a), reason for getting different dark current (even though the measured difference is in pico amp) needs to be addressed with proper explanation using appropriate experiments or literatures support. As compared with the parallel measurements perpendicular measurement are exhibiting linear dark current.

3. Energy-dependent x-ray attenuation of different materials simulated using the NIST-XCOM database shown in fig.4, y-axis unit needs attention. Instead of absorption, attenuation word may be preferred. Later in Supplementary Fig. 14 authors uses Attenuation.

4. The stopping power depends on a few parameters such energy (E) of the incident X-ray photon, high atomic number (Z), density (ρ) and thickness of the material, X-ray images of the BiOI single crystal or device are needs to be included to under the stopping power.

Thank you very much for handling our paper, and for conveying to us the comments from the Reviewers. We are delighted that Reviewers liked the novelty of the findings, and the detailed analyses we have put forward. We have now fully addressed all comments, as detailed below.

The comments from the Reviewers are in black, our replies are in blue, and all responses are numbered as R1.1, R1.2, etc.

We have added Kavya Reddy Dudipala as a co-author because of the significant contributions she made to the experiments we did in these revisions, and this has been agreed by all authors.

Finally, thank you for reminding us of the editorial policies. We have completed the editorial policy checklist, as well as the code and software submission checklist. We have already provided a data availability statement, and confirm that we will deposit all raw data into an open access repository prior to publication and include the accession code. With this revision, we have included a .zip file containing our code. This will be uploaded to the same repository as the experimental data, and they will therefore have the same accession number. We have checked the lasing reporting summary, and confirm that it is not relevant to this paper because we do not report lasing from devices or materials. We also did not perform any sex- or gender-based analyses, since this is a paper in the field of physical sciences, and involves no use of human or animal samples.

Yours Sincerely,

Robert Hoye, on behalf of all authors

Reviewer 1:

In this manuscript, Jagt et al. report the result from their systematic study of BiOI single crystals for detecting low dose rates of X-rays, which is critical for making safer radiology instruments. Through use of ultrafast spectroscopy, first-principles computations and detailed optical and structural characterisation, they obtained high sensitivity and low dose rate BiOI-based X-rays detecting device. Moreover, they discussed the underlying physical mechanism of these phenomena, and offer new design avenues for novel materials with heavy elements and low-dimensional electronic structures for (opto)electronic applications. This manuscript is excellent in terms of research conception, design, analysis and discussion of results, obtained device performance. So, I think this manuscript is a well-written article containing interesting results which merit publication.

We thank the Reviewer for their time in evaluating our paper, and for their strongly positive and constructive comments. We have now addressed all points raised, as detailed below.

1. After reading this manuscript, I think the outstanding problem is that the analysis and discussion of the results are not deep enough. The authors adopted a very classic writing style: separating the presentation of the results from the discussion. However, in the last section "Discussion", the authors actually provide more analysis of the underlying mechanism. In the previous sections of the results, the authors described the data in detail and well explained the differences between the results under different conditions. This is necessary for in-depth understanding of data and phenomena, but it will also submerge the discussion and presentation of system mechanism and new physical insights.

Our response R1.1:

We thank the Reviewer for their comment, and the opportunity to clarify how we structured this paper. This paper makes the outstanding discovery of a bismuth-halide system that can avoid carrier localisation, and provides detailed mechanistic insights into how and why this occurs, and how this can be harnessed to achieve effective radiation detectors. The rigorous, and in-depth nature of our investigations and discussion can be seen from the 35 figures we have in the Supplementary Information, which provide details that supplement the four figures we have in the main text that cover spectroscopy, structural characterisation, computations, and devices. We intentionally structured the paper to present the key findings in the main text in order to appeal to a wide audience, with further details for specialists in the Supplementary Information.

In the main text, the Results are structured to systematically cover 1) structural and spectroscopy characterisation, 2) computations, 3) in-depth experimental analyses into the delocalised photoexcited carriers in BiOI, and 4) device development. These are followed with a Discussion section that draws together our key insights and goes further to apply our learnings towards the wider field.

To illustrate the in-depth nature of our work, we will briefly cover the key findings from each section, and how they are linked together:

In the first results section, we report several unusual phenomena: i) PL with two peaks that are red-shifted to the optical bandgap, ii) PL lifetimes that increase from 2 ns at room temperature to 6.8 μ s at 80 K, iii) damped coherent oscillations from the ground state bleach decay in transient absorption spectroscopy (TAS) measurements (Fig. 1). To understand these unusual phenomena in greater depth, we analysed the TAS measurements, extracting the periods of the oscillations by taking a Fourier transform. Remarkably, we found two phonon modes which exactly matched with two of the optical modes found from Raman spectroscopy. To understand the mechanism by which these phonon modes were generated, we performed excitation-dependent TAS measurements, and found that the two phonon modes in the excited state were only generated with above bandgap photo-excitation, which showed that these were created due to impulsive absorption.

To understand these results in greater detail, we proceed to the next section focussing on first-principles computations (Fig. 2). Here, the core focus was on understanding the unusual red-shifted and broad PL peaks. Based on the TAS analysis in the

previous section, we put forward the hypothesis that the red-shifted PL spectra was due to the displacive excitation of coherent phonons instead of self-trapping. Indeed, from computations, we found that the electron and hole wavefunctions of the excitons were delocalised across each layer of BiOI, confirming that self-trapping did not occur in-plane. Furthermore, we calculated the configuration coordinate diagram for BiOI, finding the excited state and ground state converge upon each other, with a distortion that gives a local minimum in energy in the excited state. This accounts for the Stokes-shifted PL. In addition, the rise in the ground state energy to excited state energy as the system is distorted suggests that excitons can relax to the ground state and non-radiatively decay. This would then limit the PL lifetime at room temperature, and explain why cooling the lattice down would significantly increase the lifetime by depopulating the phonon modes and suppressing this non-radiative loss channel.

Following from these important mechanistic insights gained from first-principles calculations, we sought to use advanced characterisation techniques to provide experimental support and insight (please refer to Fig. 3 in the main text). We measured the Urbach energy of the crystals through transmittance measurements (coupled with detailed optical simulations to fit the data, see Supplementary Note 2). We found that we could fit the temperature-dependence of the Urbach energy with the Dow-Redfield model with a phonon energy of 18.8 meV, which matched the energy of one of the phonon modes we found from transient absorption spectroscopy (19 meV). Not only does this show us the high-quality of the crystals (since there was no sub-bandgap absorbance from defects), it verifies our earlier measurements and emphasises the strong coupling between excited-state charge-carriers and intralayer breathing modes. We also performed magneto-optical spectroscopy measurements, from which we found that the 1s exciton at 2 K has a r.m.s radius of 15.3 Å, which experimentally verifies their delocalised nature.

The delocalised nature of charge-carriers, coupled with the high atomic number and density of BiOI suggests the material to be a highly-promising candidate for radiation detection. To fully understand its potential, we measured the resistivity and mobilities of BiOI. We found the resistivities to be very high, owing to the low defect densities in these materials, as found from SCLC measurements. We also used both SCLC and time-of-flight measurements to obtain the mobilities, which substantially exceed the values of systems undergoing carrier localisation. This verifies that the material has large polarons rather than self-trapped excitons.

Finally, we apply BiOI as an X-ray detector (Fig. 4), measuring a detectable signal at dose rates as low as 22 nGy_{air} s⁻¹, substantially out-performing incumbent and other emerging technologies (as detailed in the main text on page 20). The sensitivities obtained (1100 μC Gy_{air}⁻¹ cm⁻²) are over three orders of magnitude higher than the only previous report of BiOI X-ray detectors (based on nanocrystal powders), and this was due to the low defect densities obtained in the BiOI single crystals, as well as its high density. In the Supplementary Information, we go into much more detail on the comparison between the perpendicular and parallel configurations, and the high activation energy barriers to ion migration in the perpendicular configuration (see Supplementary Note 14).

The Discussion section draws these in-depth insights together and examines the broader implications of this work on the wider field. We cover three important points:

- BiOI undergoes a structural reorganisation following photo-excitation, and is able to avoid carrier localisation. This occurs because of the coupling to two intralayer breathing modes that act perpendicular to the direction of in-plane transport. We suggest that other heavy metal halide compounds may also be found that similarly avoids carrier localisation based on structural and electronic analogy. This provides an important new direction for the field, and could lead to the design of a new generation of high-performance semiconductors;
- Drift and diffusion lifetimes can be substantially different due to the effects of charge-carrier coupling. The diffusion length of BiOI is limited because of non-radiative loss channels arising from Fröhlich coupling. Applying an electric field to decouple charge-carriers from lattice vibrational modes allows this loss channel to be suppressed at room temperature. This provides a critical new direction for groups exploring the effect of carrier-phonon coupling, motivating investigations into the effects of electric field on local changes in charge-carrier transport properties;
- We suggest that drift-driven devices are an important avenue for perovskite-inspired materials (PIMs). Many of these compounds have been found to have strong Fröhlich coupling or carrier localisation, which fundamentally limits their performance in diffusion-driven devices (e.g., photovoltaics). Our work shows that these limitations can be overcome by applying an electric field (see point above), and this motivates future efforts with PIMs towards drift-driven devices, such as radiation detectors.

To improve the clarity of these points, we have made changes to the discussion:

“Another important insight from this work is that the reorganisation of the lattice from carrier-phonon coupling can result in an unavoidable non-radiative loss channel, even if the interactions are only Fröhlich. This loss channel limits the PL lifetime in BiOI to 2 ns at room temperature, thus limiting the out-of-plane diffusion length to $\sim 0.4 \mu\text{m}$, well below the $180 \mu\text{m}$ transit distances in the perpendicular devices. The thin films used in optoelectronics will have shorter diffusion lengths, thus highlighting the limitations of using BiOI for diffusion-driven devices, such as photovoltaics. Nevertheless, we show that applying an electric field allows these carriers to be efficiently extracted. For example, applying a small bias of 1.8 V across the electrodes of the perpendicular devices produces an electric field of 100 V cm^{-1} . As a result, a drift length on the order of 1 mm would occur, thus allowing efficient charge-carrier extraction. At the same time, dark current densities remain low (1 pA mm^{-2} at 1.8 V, or 13 pA mm^{-2} at 5 V). The long drift lengths and the low dark current densities together enable the high sensitivities and low LoDDs we observed in X-ray detectors. We attribute the differences between drift and diffusion lengths to be due to the applied field causing charge-carriers to be decoupled from lattice distortions, and therefore avoid the non-radiative loss channel due to Fröhlich coupling, such that much longer drift lifetimes are obtained. Indeed, we could see from the temperature-dependent time-resolved PL measurements (Fig. 1c) that suppressing this loss channel by depopulating the dominant phonon modes allows orders of magnitude increases in the PL lifetime. This motivates a reconsideration of the applications of perovskite-inspired materials containing heavy elements (which have currently mostly focussed on photovoltaics), since we show that high-performance can still be achieved in drift-driven

devices (such as X-ray detectors), even if there is significant carrier-phonon coupling that limit diffusion lengths^{8,12-15,20,21,34,35}.”

2. In the first paragraph of Page 15, the conclusion “Calculating the effects of charge-carrier interactions with all of these modes is computationally prohibitive, and we therefore focus only the two modes that we found experimentally from TA analysis (Fig. 1d–f) to dominate coupling to charge-carriers.” needs more detailed derivation and discussion.

Our response R1.2:

Up to this point of the paper, our discussion had mostly focused on charge-carrier coupling to the two dominant phonon modes we found from transient absorption spectroscopy measurements. Having concluded this discussion, we wished to ensure that the reader is clear that there are of course many more phonon modes present in the system, as can be seen in the phonon dispersion curve shown in Fig. 2a. A ‘nice to have’ is certainly to calculate how the energy of the ground state and excited state of the lattice vary with coupling to all of the possible longitudinal optical phonon modes. However, the computational cost of systematically studying all possible couplings is prohibitive because BSE calculations are computationally expensive, and the number of possible phonon modes is very large (both in terms of frequency and momentum). This is why, in this work, the experimental identification of the two dominant coupling modes (Q_A and Q_B) is essential for the subsequent theoretical analysis: it allows us to go from a computationally intractable number of phonon modes to just two. In our view, this is a nice demonstration of the power of a joint theory-experiment work.

To improve the clarity of this point, we have modified this part of the main text as follows:

“Other phonon modes, apart from these two intralayer modes, are of course present, as shown in the phonon dispersion curve in Fig. 2a. Calculating the effects of charge-carrier interactions with all of these modes is computationally prohibitive because the number of possible phonon modes is very large, and capturing all of these phonon modes within finite differences would require very large supercells that would make the BSE calculations extremely computationally expensive. Experimentally finding from TA measurements (Fig. 1d–f) that two phonon modes dominate coupling to charge-carriers was therefore critical to making an advanced first-principles analysis of this system computationally tractable.”

3. In the determination of crystal structure, the authors grind the prepared single crystal and test it with powder XRD. Why not do single crystal XRD test directly? Isn't it possible to get more accurate results?

Our response R1.3:

The single crystals are indeed of diffraction quality, and we show our measurements of the diffraction pattern from the (00l) surface in Fig. 1a, which confirms that all peaks from this face are (00l) peaks. Performing single crystal diffraction on this material and analysing it would be heavily time consuming, and made more challenging by the fact that these crystals grow anisotropically (width/length on the order of 2-5 mm; thickness

on the order of 0.2 mm). In Supplementary Fig. 3, and later in Supplementary Fig. 6-8, we wished to verify the structure of the material and determine how the lattice parameters and atomic displacement parameters change as a function of temperature. Both can be reliably obtained from the crystals we grew by crushing into powder and performing powder diffraction. The lattice parameters, atomic positions and bond angles we determined from the crystals we grew are displayed in Supplementary Tables 1 and 2, and are fully consistent with previous detailed studies into BiOI¹⁻³. Analysis of the powder diffraction measurements is therefore adequate for this work as further information to support our work.

4. In the summarizing paragraph, the authors only summarized the results of this research work in general terms, and did not highlight their findings and innovations.

Our response R1.4:

We thank the Reviewer for pointing this out and have now edited the concluding paragraph to bring out the key findings and innovations:

*“In summary, we showed that photoexcited carriers in BiOI couple to two optical phonon **intra-layer** breathing modes. The low stiffness of the lattice along the out-of-plane (c-axis) direction, along with the intermediate coupling between carriers and longitudinal optical phonons, results in a structural deformation of the lattice along the c-axis. **The changes in the lattice energy in the excited and ground state with these distortions** causes the PL to become red-shifted, **and the PL lifetime fundamentally limited to approximately 2 ns at room temperature due to the creation of a non-radiative loss channel. We show that this limitation can be overcome by i) depopulating the dominant intra-layer phonon modes, resulting in PL lifetimes increasing to the microsecond level at 80 K, or ii) applying an electric field to decouple charge-carriers from lattice vibrations, thus enabling long drift lengths on the millimeter-scale at room temperature. Furthermore, the** excited carrier wavefunctions remain delocalised, yielding highly mobile carriers with large drift $\mu\tau$ products. The BiOI single crystals have a high electrical resistivity, and high **linear** X-ray attenuation coefficients, making them ideal X-ray detector materials. The optimised devices exhibited a high sensitivity of up to $(1.1 \pm 0.1) \times 10^3 \mu\text{C Gy}_{\text{air}}^{-1} \text{cm}^{-2}$ and a low detection limit of $22 \text{ nGy}_{\text{air}} \text{s}^{-1}$ due to the high drift $\mu\tau$ products, **and low dark current densities on the order of pA mm⁻². These findings emphasise drift-driven devices to be an important application route for perovskite-inspired materials. Importantly, our work offers structural and electronic insights into how carrier localisation could be avoided, forming a basis for the future design of perovskite-inspired materials with delocalised excitations and long charge-carrier transport lengths.”***

5. In the experimental method, the authors should provide key technical parameters.

Our response R1.5:

We have now checked through the Methods section and have ensured that key technical parameters are provided. For example, we specify the purity and supplier of the precursors, precise masses of precursors used, volumes of liquids and vacuums employed. In the characterisation section, we provide details on the excitation

wavelength, repetition rate of the lasers used and measurement time-intervals. Please note that we have not included the fluences in the Methods because different values were used in different figures. It is clearer to read the fluences we specified in the caption of the figures that relate to the specific data (e.g., see caption of Fig. 1). To make this clearer, we have signposted this to the reader in the Methods section. We have also provided full experimental details to the extra results we have added, such as the technical details on how the radiographs were taken.

For the computational section, we have included key details of the software package used, level of theory, code used, supercell size, convergence details, and many more, to ensure our work can be reproduced by other groups.

6. In the description of DFT calculation method, the authors use various calculation software and methods. Are the results reasonable and reliable?

Our response R1.6:

Indeed, we carefully designed the computations to ensure their reliability, and made several comparisons with experiment throughout the paper to ensure that they are reasonable and accurate.

To explain this in detail, we would like to first explain the calculation protocol we used. We will focus on how using this protocol yields the results that we present in the manuscript and why they are robust, considering the methods and software employed.

We started the first-principles study from the crystal structure obtained from the Materials Project database (mp-22987) and used this crystal structure to calculate the ground-state geometry. For this we employed DFT, as implemented in the electronic structure code QUANTUM ESPRESSO, using the parameters described in the Methods Section of the main text. Having calculated the optimised geometry, we subsequently calculated the vibrational properties from finite differences using QUANTUM ESPRESSO in combination with our in-house code called Caesar (for more details, see Phys. Rev. B **92**, 184301, DOI: 10.1103/PhysRevB.92.184301). Caesar creates the displaced structures that are needed to calculate the matrix of force constants with DFT, which is then Fourier transformed and diagonalised to obtain the vibrational frequencies and vectors.

Subsequently, we computed the electronic properties, for example the band structure and bandgap of BiOI, again using density functional theory within QUANTUM ESPRESSO.

DFT, as a mean-field theory, systematically underestimates the band gap of semiconductors (up to a few eV). A common approach to correct the bandgap of the studied material is to calculate the so-called quasiparticle bandgap *via* a quasiparticle correction. One widely used approach to do so is called the *GW* approximation. This approximation uses the DFT wavefunctions as a starting point, and then calculates the quasiparticle correction to the electronic eigenvalue to include the interaction of the electronic states with a screened electron cloud in the material. In other words, *GW* adds the electron-electron interaction that is neglected within the mean-field approach of DFT. In the case of BiOI, we see that the optical bandgap in DFT is widened from

1.25 eV to 2.27 eV. This very closely matches the experimental bandgap of BiOI that we measured at 4 K (approximately 2.2 eV), as we noted on page 12 of the main text:

“The calculated eigenvalues are in excellent agreement with the onset of the low-temperature absorption spectrum (~2.2 eV) in Fig. 1c.”

GW calculations are only implemented in very few electronic structure codes, one of them being YAMBO. The core functionalities of YAMBO are:

- to calculate the corrected eigenvalues from the DFT band structure using the GW approximation and
- to calculate the excitonic eigenstates from a DFT or GW wavefunctions.

YAMBO is implemented to use the DFT wavefunctions from the electronic structure codes QUANTUM ESPRESSO and ABINIT. Using the DFT wavefunction files from those, YAMBO calculates many-body corrections and excitation properties in materials, in which many-body interactions are non-negligible.

Having calculated the improved eigenvalues from the electronic states in BiOI, we then calculate the excitons using the Bethe-Salpeter equation. This sequence of electronic structure calculations is the current state-of-the-art for excitonic calculations and has been tested numerous times for many different materials [please refer to 10.1103/PhysRevLett.125.107401, <https://doi.org/10.1038/s41699-022-00355-z>, <https://doi.org/10.1021/acsnano.0c00309>], and has been described methodologically [<https://iopscience.iop.org/article/10.1088/1361-648X/ab15d0>].

The main results of the manuscript are therefore all based on the DFT wavefunctions of QUANTUM ESPRESSO, which makes them reliable regarding the software and the methods. For each step we conducted thorough convergence tests regarding the numerical parameters. We visualise the process in the flowchart below:

Supplementary Fig. 35 | Flowchart of the first-principles protocol employed in this work. The starting point of first-principles calculations is always the ground-state crystal structure, which was obtained from Materials Project. Using the crystal structure of BiOI we conducted a geometry optimisation using density functional theory as it is implemented in the open source electronic structure code QUANTUM ESPRESSO. The geometry optimisation was then followed by an electronic structure calculation, again using DFT. This gave us the mean-field electronic wavefunctions for the equilibrium structure. In parallel, we used the optimised geometry and conducted phonon calculations using finite differences with our in-house code CAESAR in conjunction with QUANTUM ESPRESSO. From the phonon calculations, we created distorted structures along the vibrational modes of interest (in this scenario the Raman-active modes observed in the TA measurement) and calculated the electronic wavefunctions for each. Then, the DFT wave functions were used for each configuration to calculate the quasiparticle corrections using the GW approximation as it is implemented in the electronic structure code YAMBO. The corrected electronic eigenvalues were used as the input to solve the Bethe-Salpeter equation to obtain the excitonic states and their eigenvalues for the equilibrium structure as well as the distorted configurations along the phonon modes studied.

For the vibrational properties we repeated the protocol, but now using the electronic structure code VASP. We did this solely to double check the robustness of the phonon-dispersion with respect to the electronic structure code and the pseudopotentials. We found that both codes (QUANTUM ESPRESSO and VASP) gave the same vibrational eigenvalues. We subsequently used these results to calculate the irreducible

representation using a post-processing tool called Phonopy. We did this because deducing the irreducible representation is not implemented in our in-house code Caesar.

The maximum displacement of atoms along the two dominant phonon modes we considered in the computations was 0.63 Å for iodine for mode Q_A , and 0.27 Å for mode Q_B for bismuth atoms. These are on the same order of magnitude as the expected thermal displacement of iodine and bismuth from refinements of the atomic displacement parameters from room-temperature X-ray diffraction measurements. Thus, the atomic displacements we considered computationally were reasonable. Indeed, we specifically covered this point in the main text on page 13:

“The maximum displacement amplitudes of Q_A , $Q_B = 10$ corresponds to the collective displacement of all iodine atoms out of the layer by $c = 0.63$ Å (Q_A), or all bismuth atoms into the layer by $c = 0.27$ Å (Q_B), which are on the same order as the expected thermal displacement of iodine and bismuth from equilibrium from X-ray diffraction measurements⁴⁵.”

To summarise, the main first-principles results presented in this manuscript were calculated with the electronic structure code QUANTUM ESPRESSO and sequentially YAMBO (which is perfectly interfaced with QUANTUM ESPRESSO) using DFT, then GW on top and BSE on top of the GW calculation. All the calculations were conducted using the same numerical parameters that have been tested rigorously.

This is a highly interdisciplinary paper, and indeed we see a core strength of this paper as synergising between synthesis, ultrafast spectroscopy, computations and detector development. We hope this paper will appeal to a broad audience. To help ensure that people from outside the theory community can understand our computational methodology and its robustness, we have included the flowchart into the supplementary information, and made reference to this figure in the Methods section.

Reviewer 2:

The author reported the use of BiOI in X-ray detection, characterized the coupling of electrons and phonons, and through first-principle calculations, systematically analyzed the process of photoexcited carriers causing lattice distortion and forming large polarons, and give the reason for the high $\mu\tau$ value of the material. In addition, a single crystal with low defect density was synthesized, and a radiation detection device with surprisingly excellent performance was made. I think the explanation of the photoexcited charge-carriers is great, but the X-ray detection performance is under debate. The current states of this paper is not suitable for publication in Nature Commun.

We thank the Reviewer for their constructive comments, and are glad that they are satisfied with the analysis of electron-phonon coupling in the photo-excited charge-carriers. We have gone through all comments, and addressed all of them, as detailed below.

1. Firstly, the PL decay time is 2 ns at room temperature, but the $\mu\tau$ product is measured as 10^{-3} cm² V⁻¹. This indicates a much higher mobility than the

measured results ($10^2 \text{ cm}^2 \text{ V}^{-1} \text{ s}^{-1}$). I think the measurements and fitting of τ requires further study. I recommend the use of alpha particles for excitation, and pulse height spectra to derive the average response amplitude.

Our response R2.1:

We thank the Reviewer for this question, which covers an important point in our work. The PL lifetimes only relate to diffusion transport. By contrast, the mobility-lifetime product obtained from the modified Hecht model relates to BiOI with a field applied (*i.e.*, drift transport).

With no applied field, photo-excited charge-carriers are coupled to the lattice and the process by which they relax to the ground state will depend on the quantity of phonons present. As discussed in the paper (with further details in Supplementary Note 4), at room temperature, a sufficient quantity of phonons are present to allow the lattice to distort extensively along a combination of phonon modes Q_A and Q_B , such that charge-carriers can then directly couple to the ground state, where the lattice then relaxes non-radiatively. This non-radiative process fundamentally limits the PL lifetime to 2 ns at room temperature.

However, when a small field is applied, this will be sufficient to separate electrons and holes from the lattice, such that they can be transported over much longer distances before undergoing recombination. In other words, the diffusion lifetime (as measured by time-resolved PL) is not the same as the drift lifetime. Thus, whilst the diffusion length of BiOI is fundamentally limited to a maximum of $0.4 \mu\text{m}$ in the out-of-plane direction at room temperature, this can be overcome by applying a field to split the electrons and holes away from vibrational modes in the lattice. For example, applying a small bias of 1.8 V across the electrodes of the perpendicular devices would give rise to an electric field of 100 V cm^{-1} , and a drift length of 1 mm. If the drift lifetimes were limited to only 2 ns, it would not have been possible to achieve millimeter-scale drift lengths, and we would not have obtained appreciable photocurrent signals through the transit distance of $180 \mu\text{m}$, let alone the 2.1 mm transit distances in the parallel devices. Thus, the fact that we obtained high photocurrent signals, and high sensitivities shows that the drift lifetimes could not have faced the same limitations due to a non-radiative loss channel as the diffusion lifetimes.

Indeed, we can see what happens to the PL lifetime when it is not fundamentally limited due to carrier-phonon coupling. In Fig. 1c, as we lower the temperature of the system to 80 K, the reduction in thermal energy results in a substantial reduction in phonon population, such that the lattice cannot distort significantly along the configuration coordinate. As a result, the excited state and ground state energies cannot come sufficiently close together for excited-state charge-carriers to directly enter into the ground state, and this non-radiative recombination channel is therefore suppressed. The PL lifetime then increases from 2 ns at room temperature to $6.8 \mu\text{s}$ at 80 K. Decoupling charge-carriers from vibrational modes in the lattice by applying an electric field would produce a similar effect.

Fig. 1 | Material properties of BiOI single crystals. c, Decay of the PL over time, measured using time-correlated single photon counting (TCSPC) from 80 K (dark blue) to room temperature (bright red).

To make this point clearer, we have modified our discussion:

“Another important insight from this work is that the reorganisation of the lattice from carrier-phonon coupling can result in an unavoidable non-radiative loss channel, even if the interactions are only Fröhlich. This loss channel limits the PL lifetime in BiOI to 2 ns at room temperature, thus limiting the out-of-plane diffusion length to $\sim 0.4 \mu\text{m}$, well below the $180 \mu\text{m}$ transit distances in the perpendicular devices. The thin films used in optoelectronics will have shorter diffusion lengths, thus highlighting the limitations of using BiOI for diffusion-driven devices, such as photovoltaics. Nevertheless, we show that applying an electric field allows these carriers to be efficiently extracted. For example, applying a small bias of 1.8 V across the electrodes of the perpendicular devices produces an electric field of 100 V cm^{-1} . As a result, a drift length on the order of 1 mm would occur, thus allowing efficient charge-carrier extraction. At the same time, dark current densities remain low (1 pA mm^{-2} at 1.8 V, or 13 pA mm^{-2} at 5 V). The long drift lengths and the low dark current densities together enable the high sensitivities and low LoDDs we observed in X-ray detectors. We attribute the differences between drift and diffusion lengths to be due to the applied field causing charge-carriers to be decoupled from lattice distortions, and therefore avoid the non-radiative loss channel due to Fröhlich coupling, such that much longer drift lifetimes are obtained. Indeed, we could see from the temperature-dependent time-resolved PL measurements (Fig. 1c) that suppressing this loss channel by depopulating the dominant phonon modes allows orders of magnitude increases in the PL lifetime.”

This finding is important not just for BiOI, but also for the wider family of polar perovskite-inspired materials. Most of the global community’s efforts with these materials have been on diffusion-driven devices (e.g., photovoltaics). Our work motivates a careful evaluation of whether carrier-phonon coupling limits the diffusion lengths, in which case, we propose that greater emphasis with these materials should be placed on drift-driven devices (e.g., radiation detectors), since an applied field can allow the separation and more efficient collection of the charge-carriers. We emphasised this point in the discussion:

“This motivates a reconsideration of the applications of perovskite-inspired materials containing heavy elements (which have currently mostly focussed on photovoltaics), since we show that high-performance can still be achieved in drift-driven devices (such as X-ray detectors), even if there is significant carrier-phonon coupling that limit diffusion lengths^{8,12-15,20,21,34,35}.”

Finally, we believe that it is more suitable to measure the mobility-lifetime product using the same X-ray source as was used for the sensitivity and LoDD measurements. As we showed in the main text in Fig. 4d, the sensitivity depends on the dose rate, and therefore will likely also be affected by the absorption profile. Alpha particles are absorbed closer to the surface than X-rays, and we believe the mobility-lifetime products obtained from these measurements would not be as comparable with the sensitivity and LoDD measurements as the mobility-lifetime values we obtained from X-ray illumination.

For the charge-carrier mobility, we obtained these values from two techniques, and both sets of measurements gave comparable values (see main text p. 18). We elaborate more on the space-charge limited current density measurements below in response to the Reviewer’s questions. For the time-of-flight measurements, the quoted mobilities of $83 \text{ cm}^2 \text{ V}^{-1} \text{ s}^{-1}$ (parallel direction) and $26 \text{ cm}^2 \text{ V}^{-1} \text{ s}^{-1}$ (perpendicular direction) were obtained under a low applied bias of 2 V, and also under low pulse fluences. Both factors ensured that no space-charge region formed, thus ensuring that the measurement conditions were as ideal as possible. We are therefore confident about the accuracy of the mobility values obtained. These values are directly relevant to the drift lengths since they were obtained in the presence of an electric field. Both sets of measurements also confirm that BiOI does not undergo carrier localisation, since the mobilities exceed $10 \text{ cm}^2 \text{ V}^{-1} \text{ s}^{-1}$.

2. In discussion part, the discussion about the limitation for diffusion-driven devices is not comprehensive. For X-ray detection, the drift length is also very long, the use of external field could not only increase the extraction efficiency, but also increase the dark current. Thus, for X-ray detection, it is also required for long carrier lifetime and diffusion length.

Our response R2.2:

We thank the Reviewer for this question, which is related to R2.1. Indeed, a long charge-carrier lifetime is required, as is a high mobility, in order to achieve a large drift length that is at least comparable to the required transit distance in order to achieve a high charge collection efficiency. We emphasise that because the application of an electric field can decouple charge-carriers from dominant phonon modes in the lattice, we can avoid the non-radiative recombination process due to carrier-phonon coupling, and therefore obtain a drift lifetime substantially longer than the diffusion lifetime (please see our detailed response above in R2.1 on this point).

In a conventional semiconductor, the unusual recombination process present in BiOI would not occur. In that case, the drift and diffusion lifetimes would be similar, and a long drift length would then also correlate with a long diffusion length. However, this

does not apply in the case of BiOI because of the differences in the recombination processes.

Regarding dark current, the low defect densities in BiOI results in high resistivities, especially in the optimal perpendicular configuration ($1.1 \times 10^{12} \Omega \text{ cm}$). The limit of detection was measured at an applied bias of 5 V (278 V cm^{-1}), and we chose this point because it is the inflection point of the photocurrent vs. applied bias curve (see Fig. 4c). At this bias, the dark current density is only 13 pA mm^{-2} , well below the dark current densities found in lead-halide perovskite single crystal detectors^{4–6}. Furthermore, we show in Supplementary Note 14 that the perpendicular devices benefit from a high activation energy to ion migration, enabling a stable dark current baseline in chopped detector measurements (see Supplementary Fig. 33).

To summarise, BiOI benefits from a low dark current density (especially in the perpendicular configuration), that remains stable and low at the comparatively low applied biases needed to obtain sufficiently long drift lengths to extract charge-carriers from the BiOI single crystal detectors.

To clarify these points, we have specifically included the following to the discussion:

“At the same time, dark current densities remain low (1 pA mm^{-2} at 1.8 V, or 13 pA mm^{-2} at 5 V). The long drift lengths and the low dark current densities together enable the high sensitivities and low LoDDs we observed in X-ray detectors.”

3. Moreover, the authors reported the perpendicular device had higher sensitivity but lower $\mu\tau$ product. This phenomenon is abnormal. The higher resistivity and lower dark current cannot determine the sensitivity, since sensitivity is caused by the carrier collection efficiency.

Our response R2.3:

We thank the Reviewer for this question, and have accordingly revisited the charge-collection efficiency of the devices and sensitivity in more detail.

The BiOI crystals grow anisotropically. Thus, whilst the parallel direction has a transport length of 2.1 mm, the perpendicular devices have a transport length of only 0.18 mm. For convenience, we have copied the illustration of the devices with dimensions below. Thus, whilst the $\mu\tau$ product is an order of magnitude smaller in the out-of-plane direction compared to in-plane, the travel distance required is also an order of magnitude smaller.

Supplementary Fig. 10 | Dark current measurements and device configurations of BiOI devices. a, Device dimensions for the parallel (left) and perpendicular (right) device configuration

In addition to the points above, the order-of-magnitude different thicknesses means that the same applied voltage will produce fields (E) an order of magnitude larger in the perpendicular devices than in the parallel devices. The net effect is that the charge collection efficiencies at the same voltage are slightly higher in the out-of-plane direction than in-plane direction.

To illustrate the points above, we can estimate the charge collection efficiencies (CCEs) using the Hecht model^{7–10}:

$$\text{CCE} = \frac{\mu\tau E}{L} \left(1 - e^{-\frac{L}{\mu\tau E}} \right)$$

In our system, we directly illuminate X-rays through one of the electrodes (in the case of the perpendicular configuration) or immediately next to one electrode (in the case of the parallel devices). This was specifically to follow the assumptions in the Hecht model as closely as practical, in which charge-carriers are generated next to one of the two electrodes. In the presence of an electric field, one of the charge carriers would be directly extracted into the adjacent illuminated electrode, while the other drifts through the length of the channel to the other electrode, depending on the polarity of the applied field. Thus, it is only the latter charge-carrier we need to consider for the overall CCE (although we note that electron and hole mobilities in this system are similar)^{7–10}. In Table R1, we summarise the $\mu\tau$ products, the electric fields a 5 V applied bias would produce, and the resulting CCEs obtained if we used the Hecht model given above. Please note that we chose an applied bias of 5 V for our illustrative example because this is the applied bias for which we reported our sensitivities and LoDDs for perpendicular devices in Fig. 4 of the main text.

Table R1. Charge collection efficiencies of the parallel and perpendicular configuration devices for 5 V applied bias

	Parallel	Perpendicular
$\mu\tau$ (cm ² V ⁻¹)	6.4×10^{-2}	1.1×10^{-3}
Channel length (cm)	0.21	0.018
Electric field (V/cm)	23.8	278
CCE (%)	93.4	97.1

To understand this further, we can see that $\mu\tau E L^{-1}$ equals $\mu\tau V L^{-2}$. Thus, for the same applied bias, we can compare $\mu\tau L^{-2}$, which are 1.45 V⁻¹ (parallel) and 3.40 V⁻¹ (perpendicular). $1 - \exp(-[\mu\tau V L^{-2}]^{-1})$ is then equal to 12.9×10^{-2} (parallel) and 5.7×10^{-2}

(perpendicular). Thus, the product between $\mu\tau L^{-2}$, $1-\exp(-[\mu\tau V L^{-2}]^{-1})$ and the applied bias V produces an overall CCE where the differences in magnitude between these two terms cancel out, producing CCE values that are very similar between the two configurations, but overall slightly higher for the perpendicular devices.

In the main text, we focus on the perpendicular devices because these provide the best demonstration of the performance and impact of BiOI for radiation detection. But in order to clarify the points made above, we have added details on the comparison between the parallel and perpendicular devices into the Supplementary Information, immediately below the photocurrent measurements for the parallel devices in Supplementary Fig. 32.

In the parallel configuration, the $\mu\tau$ product is an order of magnitude larger than in the perpendicular configuration ($1.1 \times 10^{-3} \text{ cm}^2 \text{ V}^{-1}$, see main text). However, we found the sensitivity to be higher in the perpendicular configuration at the same applied bias. This is because despite the order-of-magnitude higher $\mu\tau$ products in the parallel configuration, the transit distances were also an order of magnitude larger. The same applied bias would also produce an order of magnitude smaller electric field. Overall, these effects cancel out between the two configurations, as illustrated in Supplementary Table 4 for the case of 5 V bias (at which we measured sensitivities and LoDDs).

Supplementary Table 4 | Charge collection efficiencies of the parallel and perpendicular configuration devices for 5 V applied bias, estimated using the Hecht model

	Parallel	Perpendicular
$\mu\tau \text{ (cm}^2 \text{ V}^{-1}\text{)}$	6.4×10^{-2}	1.1×10^{-3}
Channel length (cm)	0.21	0.018
Electric field (V/cm)	23.8	278
CCE (%)	93.4	97.1

To understand the illustration in Supplementary Table 4 more intuitively, we can examine the Hecht model:

$$\text{CCE} = \frac{\mu\tau E}{L} \left(1 - e^{-\frac{L}{\mu\tau E}} \right)$$

where CCE is the charge-collection efficiency, E the applied field (V L^{-1}) and L the transit length (which is taken to be the same as the distance between the electrodes). This applies to the case where electrons and holes are generated next to one electrode, and only one charge-carrier is transited to the other electrode. We can see that $\mu\tau E L^{-1}$ equals $\mu\tau V L^{-2}$. Thus, for the same applied bias, we can compare $\mu\tau L^{-2}$, which are 1.45 V^{-1} (parallel) and 3.40 V^{-1} (perpendicular). $1-\exp(-[\mu\tau V L^{-2}]^{-1})$ is then equal to 12.9×10^{-2} (parallel) and 5.7×10^{-2} (perpendicular). Thus, the product between $\mu\tau L^{-2}$, the applied bias V , and $1-\exp(-[\mu\tau V L^{-2}]^{-1})$ produces an overall CCE where the differences in these terms cancel out, producing CCE values that are very similar between the two configurations, with the perpendicular configuration overall having a slightly higher CCE, which would lead to higher sensitivities. Furthermore, the perpendicular configuration is a more ideal structure than the parallel configuration for radiation detector measurements because the active area is more precisely defined, and the dose rate of X-rays reaching the active part of the device can be more accurately calculated. We

therefore only report the sensitivities and LoDDs for the perpendicular devices in this work.

In addition, we have made the following modifications to the main text on page 20:

“Similar behaviour was found in the parallel configuration, and this detailed in Supplementary Note 14, from which we obtained a higher mobility-lifetime product of $(6\pm 2) \times 10^{-2} \text{ cm}^2 \text{ V}^{-1}$. Nevertheless, devices made in the perpendicular configuration have higher charge-collection efficiencies for the same applied bias, due to the shorter transport distances and higher electric fields (see Supplementary Table 4). The sensitivities of $(1.1 \pm 0.1) \times 10^3 \mu\text{C Gy}_{\text{air}}^{-1} \text{ cm}^{-2}$ (Fig. 4e) reached in the perpendicular configuration substantially exceeds the performance of previously-reported nanocrystalline BiOI detectors¹¹.”

Other questions:

1. In Supplementary Fig. 14g, I suggest that the author use a logarithmic coordinate system instead of the linear coordinate system, just like Fig 1c. Only attenuation to 20% cannot effectively judge the fitting value of the lifetime. At the same time, it is better to choose a longer range of abscissa. Furthermore, it seems that the wavelength of 400 nm has a longer life under excitation, and I'm curious about the reason.

Our response R2.4:

We have checked Supplementary Fig. 4g, and confirm that the vertical axis of the time-resolved PL plot is on a logarithmic scale. At the same time, we recognise that the way we originally presented this data did not make this clear. We have therefore changed the limits of the vertical axis and replotted it, as copied below. Furthermore, we have extended the time range on the abscissa to 1000 ns, showing the decay in the normalised PL by more than an order of magnitude.

Supplementary Fig. 4 | Photoluminescence from BiOI single crystals. g, Normalised time-resolved PL of a BiOI single crystal at 4 K, measured with an intensified charge-coupled device (iCCD) detector, and illuminated with a pulsed 400 nm and 530 nm laser. The repetition rate of the laser was 1000 Hz. Both series depict similar decay profiles. This indicates that there is no to little effect caused by surface recombination at the crystal surface.

The reason we put this data in was to determine whether the ultrafast spectroscopy measurements performed under a blue wavelength excitation were reflective of the properties of the bulk. We therefore compared the PL decay of BiOI when photo-

excited at a longer wavelength of 530 nm (closer to the bandgap at 640 nm wavelength). Supplementary Fig. 4g presents a zoom in to the initial decay (bearing in mind that the PL decays over a $>5 \mu\text{s}$ timescale), where any surface recombination effects are typically manifest.

From this comparison, the two PL decays were very similar. One may judge the PL decay with 530 nm wavelength excitation to be very slightly slower, however, the difference in normalised PL signal was only 8% of the average signal between the two traces at the tail of this decay. By contrast, if we take the mean difference in normalised PL over the entire trace, it is 20%. Thus, any differences in the PL decays of the two traces were within uncertainty. We would therefore conclude that the measurements made with 400 nm wavelength excitation were reflective of the bulk photophysical properties of BiOI. This was because we were photo-exciting on the (00l) face of the BiOI crystal. Due to the layered nature of the structure, this face was not terminated with broken dangling bonds, and it is therefore reasonable that we would not have significant surface recombination on this surface.

Indeed, we wrote on page 9 of the main text that:

“Finally, we note that we compared the PL decay of BiOI crystals with 400 nm and 530 nm wavelength excitation, and found no significant change in the PL lifetime (Supplementary Fig. 4g). This observation suggests that there are no dangling bonds that potentially create surface in-gap states at the probed (00l) surface (see diagrams in Supplementary Fig. 1), and that the measurements made and discussed here are reflective of the bulk properties of BiOI.”

2. I have some confusion about the carrier mobility tests.

(1) In Supplementary Fig. 12, the ohmic zone should be larger, including the part with a smaller slope in the TRAP-SCLC zone. In my opinion, their slopes are very close to 1. This is in sharp contrast to the part with a slope far greater than 1.

Our response R2.5:

We thank the Reviewer for this question. The slope on a log-log scale in the region labelled Trap-SCLC is 1.4, and therefore deviates from the initial linear region that would clearly be part of the Ohmic regime (where the slope is exactly 1). This can occur if there are a distribution of shallow trap states¹²⁻¹⁴, but we accept that other factors could play a role. Nevertheless, we can clearly see a trap filling regime, and a trap-free space-charge regime (where there is a quadratic dependence of current on voltage). These latter two regimes are the ones which are important for us to determine trap density and SCLC mobility.

We have modified the wording in the caption of Supplementary Fig. 12:

“In these dark current measurements, there is a clear initial Ohmic regime, before entering into a space-charge regime after going through a non-linear regime (which can be due to the effects of shallow traps), and a steep trap filling regime.”

(2) Why the SCLC test is only carried out in one direction? It is suggested that the experiment in the vertical direction can be supplemented.

Our response R2.6:

Indeed, it would be ideal to compare the SCLC mobilities from both the parallel and perpendicular directions. However, in our attempts at performing SCLC measurements on the perpendicular devices, we were not able to enter into the space-charge regime. By contrast, our parallel configuration measurements shown clearly have a space-charge region and trap filling regime, and we could reliably obtain a mobility value. The mobility obtained ($54.3 \text{ cm}^2 \text{ V}^{-1} \text{ s}^{-1}$) is reasonably similar to the value obtained from time-of-flight measurements ($83 \text{ cm}^2 \text{ V}^{-1} \text{ s}^{-1}$), if we consider the fact that these measurements were made in completely different ways. From time-of-flight measurements, we were able to obtain a mobility for the out-of-plane direction ($26 \text{ cm}^2 \text{ V}^{-1} \text{ s}^{-1}$), which is reasonable and consistent with the lower band dispersion between planes.

To address this point, we have added the following to the caption of Supplementary Fig. 12:

“Please note that SCLC measurements of perpendicular BiOI devices are not shown because these did not enter into a clear space-charge regime.”

(3) When the resistivity differs by three orders of magnitude, the mobility of parallel electrodes and vertical electrodes is similar. This doesn't seem very reasonable.

Our response R2.7:

The cause of the large difference in resistivities is the differences in ion migration between the parallel and perpendicular configurations. As detailed in Supplementary Note 14, ion migration occurs more easily in the in-plane than the out-of-plane direction in BiOI. As seen in Supplementary Fig. 33a (copied below), there is mixed ionic and electronic transport in the in-plane direction, and both effects therefore contribute to the resistivity in the parallel direction. By contrast, in the perpendicular direction, ionic effects are substantially reduced (Supplementary Fig. 33c), resulting in orders of magnitude larger resistivities.

To clarify this point, we have made the following change to page 17 of the main text:

“The resistivity in the perpendicular configuration is orders of magnitude larger than the parallel configuration because of suppressed ion migration (see Supplementary Fig. 33 and 34). That is, in the in-plane direction, mixed ionic and electronic transport occurs to a much larger extent than in the out-of-plane direction, resulting in an orders of magnitude difference in resistivity despite the mobilities in the two directions being similar. Notably, the resistivities

obtained from BiOI in the perpendicular configuration is orders of magnitude larger than the resistivities of recently-reported novel bismuth-halide semiconductors (e.g., $\text{MA}_3\text{Bi}_2\text{I}_9$; 10^{10} – $10^{11} \Omega \text{ cm}$)^{15,16}, CZT ($10^{10} \Omega \text{ cm}$)¹⁶ and 3D perovskites (10^7 – $10^8 \Omega \text{ cm}$)¹⁶.”

Supplementary Figure 33 | Stability against ionic drift. Current vs. time traces of BiOI devices in the **a**, medium-field parallel (133 V cm^{-1} applied field; 30 V bias), **b**, medium-field perpendicular (111 V cm^{-1} applied field; 2 V bias), and **c**, high-field perpendicular (1667 V cm^{-1} , 30 V bias) configurations with and without X-ray illumination. From these measurements, it can be seen that there was ionic drift in the parallel devices only, given that the baseline current changed over time, whereas the perpendicular device baseline current remained steady over time.

(4) In Fig. 3d and Fig. 3e, the inflection point is blocked by the fitting point. It is suggested that these lines can be separated. If the inflection point is unclear, you can add auxiliary lines or supplement the enlarged picture. It is not recommended to use solid big dots for marker symbols.

Our response R2.8:

This is certainly a good point. We have modified Fig. 3d,e to remove the large points and add in small auxiliary lines to show how we arrived at the inflection points. We

have also added the data as larger figures to the Supplementary Information, so that the data can be more easily seen.

Fig. 3 | Photophysical properties in BiOI. d,e, Normalised transient current curves of the BiOI single crystal device under various biases for the perpendicular (d) and the parallel structure (e). Tangents fit to obtain the arrival times are indicated. Please refer to Supplementary Fig. 14 for more detailed views of these tangents.

Supplementary Fig. 14 | Time-of-flight measurements for BiOI single crystal devices. Individual plots of the transient current curves in the (a – c) perpendicular, and (d – f) parallel configurations under different applied biases shown inset. Tangents fit, and their intersections to determine the arrival time are indicated in dashed lines.

The numbers of all subsequent supplementary figures have been updated.

3. The authors mentioned that the distortion of the lattice could be controlled by material design. Among the materials with broad and Stokes-shifted PL peak, can you give readers more suggestions through first-principle calculations, and screen out materials with the nature of delocalization instead of self-trapping?

Our response R2.9:

This is an excellent question by the Reviewer, and exactly the question we wish the community to be thinking about after reading our paper. The findings in this work on a materials systems (BiOI) that can avoid carrier localisation is highly novel. Currently, the wider field believes that small polaron formation is a hallmark of bismuth-halide materials. This work identifies BiOI to be an exception, in that the photophysical properties can be fully explained by Fröhlich interactions rather than small polarons or self-trapped excitons. This enables high mobilities substantially exceeding those of a system with carrier localisation (which would be $<10 \text{ cm}^2 \text{ V}^{-1} \text{ s}^{-1}$, sometimes $<1 \text{ cm}^2 \text{ V}^{-1} \text{ s}^{-1}$)^{17–20}, and this is critical for effective X-ray detector performance. The bulk of this paper is focussed on understanding this phenomenon in BiOI through detailed spectroscopic measurements, magneto-optical spectroscopy, electronic characterisation and advanced first principles calculations. At the end, in the Discussion section, we offer our views on some of the parameters that enabled the delocalised nature of charge-carriers found here, which may act as a guide to finding other compounds with similar properties:

“We show that this structural reorganisation of the lattice is an intrinsic property of the material, rather than being induced by crystal defects. This suggests that the structural deformation can be controlled by strategic materials design (e.g., by changing the number of layers or composition), which can lead to further compounds with similar desirable properties. In contrast to typical van der Waals materials, each layer in BiOI is comprised of a larger number of atoms through the cross-section (i.e., I–O–Bi–O–I), which results in the material exhibiting two breathing modes (instead of one) that modify the excited state. We propose that layered ionic materials (where the charge densities of the electrons and holes are fully symmetric and entirely delocalised in the in-plane directions) composed of heavy atoms and exhibiting multiple breathing modes (i.e., contain multiple layers), will exhibit similar properties to BiOI. In principle, the symmetry and delocalisation of the wavefunctions can be easily checked by examining the projected density of states of candidate materials, in which the conduction band is primarily composed of states of the inner atoms (e.g., Bi and O) and the valence band primarily outer atom states (e.g., I).”

These perspectives can form the basis for the development of design rules to pinpoint heavy pnictogen-halide compounds that can avoid carrier localisation, which will be important not just for radiation detectors, but more broadly in energy devices (such as indoor photovoltaics or photocathodes for water splitting). Systematic, in-depth investigations into several materials systems based on our perspectives will be required to develop and refine a set of design rules, which goes well beyond the scope of this work. As an intermediate step, one could screen through several compounds based on the deformation potential, which is one the key parameters determining whether carrier localisation occurs due to strong electron coupling to acoustic

phonons. But this effort in itself would be a separate paper, and we are indeed working on this.

We hope that our work will inspire the wider field to join us in this endeavour, and to capture this we have added the following to the end of the concluding paragraph:

“Importantly, our work offers structural and electronic insights into how carrier localisation could be avoided, forming a basis for the future design of perovskite-inspired materials with delocalised excitations and long charge-carrier transport lengths.”

Reviewer 3:

Manuscript titled "Layered BiOI single crystals capable of detecting low dose rates of X-rays" discusses performance of the device with appreciable sensitivities reaching $1.1 \times 10^3 \mu\text{C Gy}^{-1} \text{cm}^{-2}$ are achieved, and the lowest dose rate directly measured by the detectors was 22 nGy^{-1} . Experimental findings discussed in this manuscript is worth for publish it in nature communication because of the novelty, systematic analysis and interpretations.

We thank the Reviewer for their strongly positive comments, especially on the novelty of this work, and the systematic nature of the analyses.

Queries to Authors:

1. What happen to the attenuation value of BiOI if the single crystal replaced with compacted nanocrystalline BiOI based pellet?

[Response redacted]

2. In supplementary fig 32 (a), reason for getting different dark current (even though the measured difference is in pico amp) needs to be addressed with proper explanation using appropriate experiments or literatures support. As compared with the parallel measurements perpendicular measurement are exhibiting linear dark current.

Our response R3.2:

We thank the Reviewer for their attention to detail (Supplementary Fig. 32, now renumbered to 33, copied below for convenience). The measurements in Suppl. Fig. 33a are of the current from the parallel configuration BiOI devices under chopped X-ray illumination with an applied field of 133 V cm^{-1} . Thus, the current changes from dark current to dark current + photocurrent. The increase in the baseline is due to ion migration causing an increase in the dark current over time. This dark current drift occurs due to ions accumulating at the electrodes and increasing the dark current injected into the device. Dark current drift due to ion migration has been widely observed in lead-halide perovskites and bismuth-based radiation detector materials²¹⁻²⁴.

As shown in Supplementary Fig. 34, the activation energy barrier to ion migration in the perpendicular devices is larger, exceeding those of lead-halide perovskites. As a result, ion migration plays an insignificant role on the dark currents under low (111 V

cm^{-1}) and high (1667 V cm^{-1}) fields, and the baseline current then remains constant over the course of the measurement.^{21,22}

Supplementary Figure 33 | Stability against ionic drift. Current vs. time traces of BiOI devices in the **a**, medium-field parallel (133 V cm^{-1} applied field; 30 V bias), **b**, medium-field perpendicular (111 V cm^{-1} applied field; 2 V bias), and **c**, high-field perpendicular (1667 V cm^{-1} , 30 V bias) configurations with and without X-ray illumination. From these measurements, it can be seen that there was ionic drift in the parallel devices only, given that the baseline current changed over time, whereas the perpendicular device baseline current remained steady over time.

To clarify this point, we have added the following to the Supplementary Information:

“We note that another important advantage of the perpendicular configuration was that we could bias these devices up to 30 V (1667 V cm^{-1}) without obtaining any baseline dark current drift, as shown in Supplementary Fig. 33c. By contrast, we found that the parallel devices started to exhibit dark current drift after applying 10 V bias (44 V cm^{-1}), as shown from the increase in the baseline dark current in Supplementary Fig. 33a. Dark current drift occurs due to ion migration in the bulk of the X-ray attenuation material¹⁸. To quantify the activation energy barrier to ion migration, we used a probe station to measure current at applied biases of 0 V and 7 V , with a biasing period of 80 s at each voltage.”

3. Energy-dependent x-ray attenuation of different materials simulated using the NIST-XCOM database shown in fig.4, y-axis unit needs attention. Instead of absorption, attenuation word may be preferred. Later in Supplementary Fig. 14 authors uses Attenuation.

Our response R3.3:

We thank the Reviewer for this comment. In Fig. 4a, we calculate the linear attenuation coefficient (α) of the materials, as per Beer-Lambert law:^{7,23}

$$I = I_0 \exp(-\alpha x)$$

Related to this is the mass attenuation coefficient (μ), which is the ratio of the linear attenuation coefficient to the density of the material, i.e., $\mu = \alpha/\rho$

From the NIST-XCOM database, we obtained the mass attenuation coefficient ($\text{cm}^2 \text{g}^{-1}$). We multiplied this by the density of all materials to obtain the linear attenuation coefficient in order to determine how strong each material would absorb ionising radiation per unit thickness, which we use as a gauge for the stopping power. This is the conventional metric groups typically use to compare radiation detector materials.²³

For the vertical axis of Supplementary Fig. 15, we have double checked in the radiography literature. What we meant to say was “absorption efficiency”, that is the percentage of the incident X-ray flux that is absorbed in the X-ray detector material as a function of material thickness.²⁵

We have therefore modified the vertical axis of Fig. 4a to “Linear Attenuation Coefficient (cm^{-1})” and Supplementary Fig. 15 to “Absorption Efficiency (%)”

We have gone through the entire main text and supplementary information to ensure our terminology is consistent.

4. The stopping power depends on a few parameters such energy (E) of the incident X-ray photon, high atomic number (Z), density (ρ) and thickness of the material, X-ray images of the BiOI single crystal or device are needs to be included to under the stopping power.

Our response R3.4:

Thank you to the Reviewer for this suggestion. We have accordingly taken radiographs of BiOI single crystals side-by-side with a silicon reference. We stacked the BiOI crystals together to obtain the same thickness (0.4 mm) as the silicon reference in order to obtain a fair comparison. This data is now added to Fig. 4a as an inset. We also quantified the transmittance through each material (Figure R2), and obtained values of 78% for Si, and 2% for BiOI. From the radiographs and transmittance measurements, it can be clearly seen that BiOI attenuates X-rays more strongly, demonstrating its higher stopping power.

Fig 4 | X-ray detection properties of BiOI single crystals and performance of devices in the perpendicular configuration. a, Linear attenuation coefficients of various compounds as a function of photon energy. **Inset are radiographs of Si (0.4 mm thick) and BiOI crystals stacked together to give the same thickness of 0.4 mm. The transmittance through Si is 78%, whereas through BiOI, it is 2%, demonstrating the high stopping power of BiOI.**

Figure R2. Measurement of the transmittance of 40 keV X-rays through 0.4 mm thick silicon and BiOI

We have added the following to page 19 of the main text:

“We took radiographs of BiOI single crystals in a side-by-side comparison with silicon, and showed that far fewer X-rays were transmitted (only 2%, compared to 78% for silicon), demonstrating the higher stopping power of BiOI (Fig. 4a, inset).”

The experimental parameters have been added to the Methods section:

“Radiographs of the single crystals were taken using a Zeiss Versa 610 X-ray computed tomography (CT) scanner, using 40 keV X-rays, with a 3 W power source. The exposure time was 120 s.”

References

1. Schultz, P. & Keller, E. Strong positive and negative deviations from Vegard's rule: X-ray powder investigations of the three quasi-binary phase systems BiOX-BiOY (X,Y = Cl, Br, I). *Acta Crystallogr. B* **70**, 372–378 (2014).
2. Dong, X.-D., Yao, G.-Y., Liu, Q.-L., Zhao, Q.-M. & Zhao, Z.-Y. Spontaneous Polarization Effect and Photocatalytic Activity of Layered Compound of BiOIO₃. *Inorg. Chem.* **58**, 15344–15353 (2019).
3. Bannister, F. A. The crystal-structure of the bismuth oxyhalides. *Mineral. Mag. J. Mineral. Soc.* **24**, 49–58 (1935).
4. Tie, S. *et al.* Robust Fabrication of Hybrid Lead-Free Perovskite Pellets for Stable X-ray Detectors with Low Detection Limit. *Adv. Mater.* **32**, 2001981 (2020).
5. Deumel, S. *et al.* High-sensitivity high-resolution X-ray imaging with soft-sintered metal halide perovskites. *Nat. Electron.* **4**, 681–688 (2021).
6. Jiang, J. *et al.* Synergistic strain engineering of perovskite single crystals for highly stable and sensitive X-ray detectors with low-bias imaging and monitoring. *Nat. Photonics* **16**, 575–581 (2022).
7. Jana, A. *et al.* Perovskite: Scintillators, direct detectors, and X-ray imagers. *Mater. Today* **55**, 110–136 (2022).
8. Kasap, S. O., Kabir, M. Z., Ramaswami, K. O., Johanson, R. E. & Curry, R. J. Charge collection efficiency in the presence of non-uniform carrier drift mobilities and lifetimes in photoconductive detectors. *J. Appl. Phys.* **128**, 124501 (2020).
9. Hecht, K. Zum Mechanismus des lichtelektrischen Primärstromes in isolierenden Kristallen. *Zeitschrift für Phys.* **77**, 235–245 (2000).
10. S O Kasap. X-ray sensitivity of photoconductors: application to stabilized a-Se. *J. Phys. D. Appl. Phys.* **33**, 2853 (2000).
11. Praveenkumar, P., Venkatasubbu, G. D., Thangadurai, P. & Prakash, T. Nanocrystalline bismuth oxyiodides thick films for X-ray detector. *Mater. Sci. Semicond. Process.* **104**, 104686 (2019).
12. Zhang, P. *et al.* Space-charge limited current in nanodiodes: Ballistic, collisional, and dynamical effects. *J. Appl. Phys.* **129**, (2021).
13. Moiz, S. A., Khan, I. A., Younis, W. A. & Karimov, K. S. Space Charge–Limited Current Model for Polymers. in (ed. Yilmaz, F.) (IntechOpen, 2016). doi:10.5772/63527.
14. Stallinga, P. *Electrical Characterization of Organic Electronic Materials and Devices.* (John Wiley & Sons, Ltd, 2009).
15. Zhuang, R. *et al.* Highly sensitive X-ray detector made of layered perovskite-like (NH₄)₃Bi₂I₉ single crystal with anisotropic response. *Nat. Photonics* **13**, 602–608 (2019).
16. Zheng, X. *et al.* Ultrasensitive and stable X-ray detection using zero-

- dimensional lead-free perovskites. *J. Energy Chem.* **49**, 299–306 (2020).
17. Pan, W. *et al.* Cs₂AgBiBr₆ single-crystal X-ray detectors with a low detection limit. *Nat. Photonics* **11**, 726–732 (2017).
 18. Huang, Y.-T. *et al.* Strong Absorption and Ultrafast Localisation in NaBiS₂ Nanocrystals with Microsecond Charge-Carrier Lifetimes. *Nat. Commun.* **13**, 4960 (2022).
 19. Wu, B. *et al.* Strong self-trapping by deformation potential limits photovoltaic performance in bismuth double perovskite. *Sci. Adv.* **7**, eabd3160 (2021).
 20. Buizza, L. R. V *et al.* Charge-Carrier Mobility and Localization in Semiconducting Cu₂AgBiI₆ for Photovoltaic Applications. *ACS Energy Lett.* **6**, 1729–1739 (2021).
 21. Yang, B. *et al.* Heteroepitaxial passivation of Cs₂AgBiBr₆ wafers with suppressed ionic migration for X-ray imaging. *Nat. Commun.* **10**, 1989 (2019).
 22. Zhang, P. *et al.* Ultrasensitive and Robust 120 keV Hard X-Ray Imaging Detector based on Mixed-Halide Perovskite CsPbBr_{3-n}I_n Single Crystals. *Adv. Mater.* **34**, (2022).
 23. Wei, H. & Huang, J. Halide lead perovskites for ionizing radiation detection. *Nat. Commun.* **10**, 1066 (2019).
 24. García-Batlle, M. *et al.* Coupling between Ion Drift and Kinetics of Electronic Current Transients in MAPbBr₃ Single Crystals. *ACS Energy Lett.* **7**, 946–951 (2022).
 25. Sprawls, P. *Screen/Film Radiographic Receptors*. (Medical Physics Publishing).

REVIEWERS' COMMENTS

Reviewer #1 (Remarks to the Author):

The authors have carefully considered the reviewers' comments and suggestions, and have made serious changes to this manuscript. I believe that the publication requirements have been met.

Reviewer #2 (Remarks to the Author):

The authors have well addressed all the questions raised by the reviewers. I think it is acceptable in current form.

Reviewer #3 (Remarks to the Author):

Authors,

I am satisfied with yours reply for my queries. Hence, myself recommending your manuscript for publication in "Nature communications".

Point-by-point response to reviewer comments:

Reviewer 1:

The authors have carefully considered the reviewers' comments and suggestions, and have made serious changes to this manuscript. I believe that the publication requirements have been met.

Reviewer 2:

The authors have well addressed all the questions raised by the reviewers. I think it is acceptable in current form.

Reviewer 3:

Authors,

I am satisfied with yours reply for my queries. Hence, myself recommending your manuscript for publication in "Nature communications".

We are delighted that all Reviewers are satisfied with the changes made and consider the paper suitable for publication in *Nature Communications*. We would like to thank the Reviewers for their time, and their care in evaluating our paper.